# Meta-learning families of plasticity rules in recurrent spiking networks using simulation-based inference

**Basile Confavreux\***
Institute of Science and Technology Austria
basile.confavreux@gmail.com

**Poornima Ramesh\***
University of Tübingen, Germany
poornimaramesh1995@gmail.com

**Pedro J. Gonçalves**
University of Tübingen, Germany
VIB-Neuroelectronics Research Flanders (NERF), Belgium
imec, Belgium

**Jakob H. Macke †**
University of Tübingen, Germany
Max Planck Institute for
Intelligent Systems, Germany

**Tim P. Vogels †**
Institute of Science
and Technology Austria

## Abstract

There is substantial experimental evidence that learning- and memory-related behaviours rely on local synaptic changes, but the search for distinct plasticity rules has been driven by human intuition, with limited success for multiple, co-active plasticity rules in biological networks. More recently, automated meta-learning approaches have been used in simplified settings, such as rate networks and small feed-forward spiking networks. Here, we develop a simulation-based inference (SBI) method for sequentially filtering plasticity rules through an increasingly fine mesh of constraints that can be modified on-the-fly. This method, *filter SBI*, allows us to infer entire families of complex and co-active plasticity rules in spiking networks. We first consider flexibly parameterized doublet (Hebbian) rules, and find that the set of inferred rules contains solutions that extend and refine—and also reject—predictions from mean-field theory. Next, we expand the search space of plasticity rules by modelling them as multi-layer perceptrons that combine several plasticity-relevant factors, such as weight, voltage, triplets and co-dependency. Out of the millions of possible rules, we identify thousands of unique rule combinations that satisfy biological constraints like plausible activity and weight dynamics. They can be used as a starting point for further investigations into specific network computations, and already suggest refinements and predictions for classical experimental approaches on plasticity. This flexible approach for principled exploration of complex plasticity rules in large recurrent spiking networks presents the most advanced search tool to date for enabling robust predictions and deep insights into the plasticity mechanisms underlying brain function.

## 1 Introduction and related work

Synaptic plasticity has received continuous attention over the past decades [1–5]. Nevertheless, how functions such as learning and memory emerge from local synaptic changes is still poorly understood. Given the experimental inaccessibility of synapses *in vivo*, many studies have relied

---

\*, †: equal contributions
37th Conference on Neural Information Processing Systems (NeurIPS 2023)

on theoretical methods to shed light on the plasticity rules at play [6–12]. However, this approach does not scale [3, 4]—in particular, previous attempts at tuning complex, co-active plasticity rules have been limited by human intuition, and had to rely on theoretical frameworks based on strongly simplifying assumptions [8, 9].

Recently, several studies have aimed to address these issues by automating the discovery of plasticity rules using numerical meta-learning approaches [13–20]. Given the non-differentiability of biological systems, along with the steep compute requirements of simulating plastic networks at scale, these studies have been restricted to rate networks [13–16, 18–20] or small feed-forward spiking networks [15, 17]. Moreover, these approaches require the *a priori* definition and optimization of a loss function that enforces a desired network computation, as well as various regularizers to ensure the implementation of the task is biologically plausible. Lastly, the above studies typically propose *single* plasticity mechanisms compatible with the data or network function considered.

Here, we aim at a more comprehensive approach to discovering biologically plausible plasticity rules. We study network dynamics in large recurrent spiking networks with plasticity rules that are flexibly parameterized (using polynomials or neural networks, Fig. 1A). We introduce filter simulation-based inference (fSBI), a new meta-learning approach that can thoroughly explore the space of potential rules. This method allows us to select sets of rules using metrics that effectively constrain network dynamics to fulfill the desired conditions. Our approach successfully infers entire families of rules that robustly establish plausible dynamics in large spiking networks. We demonstrate the scalability of fSBI by applying it to neural-network-based search spaces that include additional factors known to influence plasticity, such as, e.g., synaptic weight or membrane potentials. Despite the nonlinear interactions induced by the neural network parameterization of plasticity and the increase in the number of factors, fSBI successfully recovers plausible rules. Interestingly, when considered in isolation, the inferred candidate rules often result in contradictory experimental predictions, revealing that the classical pre-post protocols may only have marginal explanatory power.

Overall, our study introduces a principled approach for meta-learning plasticity rules in complex networks. The inferred distributions of plausible rules can be used as starting points for future studies of specific network computations, and raise questions about classical theoretical and experimental protocols for understanding plasticity.

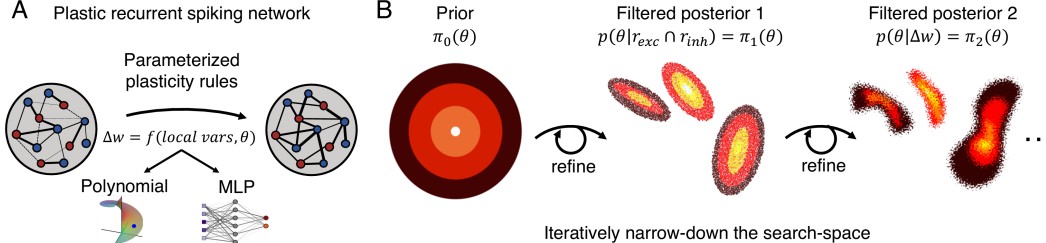

Figure 1: *Flexible inference of plasticity rules in recurrent spiking networks with fSBI.*
**A**: Plastic recurrent spiking networks are simulated with parameterized plasticity rules (parameters $\theta$), either as polynomials (Figs. 2, 3) or as multi-layer perceptrons (Fig. 4). **B**: Filter-SBI (fSBI). Starting from a prior over plasticity parameters $\pi_0(\theta)$, we infer a posterior given a first network metric, for instance the population firing rate, $p_1(\theta|r_{\text{exc}} \cap r_{\text{inh}})$. In contrast to other classical Bayesian inference approaches, $\pi_1 = p_1(\theta|r_{\text{exc}} \cap r_{\text{inh}})$ is used as a prior for the inference in the subsequent round, given a *different* network metric. We then repeat this process for various network metrics.

## 2 Methods

### 2.1 Spiking network model

We consider a recurrent spiking network of 4096 excitatory and 1024 inhibitory leaky integrate-and-fire neurons with conductance-based synapses, relative refractoriness, adaptive threshold and NMDA currents. The model parameter values follow Zenke et al. [8] (Fig. 1A; see Supp. 6.1 for a complete description of parameters). The membrane potential dynamics of neuron $j$ (excitatory or inhibitory) are given by

$$\tau_m \frac{\mathrm{d}V_j}{\mathrm{d}t} = -(V_j - V_{\text{rest}}) - g_j^{\text{E}}(t)(V_j - E_{\text{E}}) - g_j^{\text{I}}(t)(V_j - E_{\text{I}}),$$ (1)

where E stands for excitation and I for inhibition. A postsynaptic spike is emitted whenever the membrane potential $V_j(t)$ crosses a threshold $V_j^{\text{th}}(t)$, with an instantaneous reset to $V_{\text{reset}}$. This threshold $V_j^{\text{th}}(t)$ is incremented by $V_{\text{spike}}^{\text{th}}$ every time neuron $j$ spikes and otherwise decays following

$$\tau_{\text{th}} \frac{\mathrm{d}V_j^{\text{th}}}{\mathrm{d}t} = V_{\text{base}}^{\text{th}} - V_j^{\text{th}}. \tag{2}$$

The excitatory and inhibitory conductances, $g^{\text{E}}$ and $g^{\text{I}}$ evolve such that

$$g_j^{\text{E}}(t) = a g_j^{\text{AMPA}}(t) + (1-a) g_j^{\text{NMDA}}(t) \quad \text{and} \quad \frac{\mathrm{d}g_j^{\text{I}}}{\mathrm{d}t} = -\frac{g_j^{\text{I}}}{\tau_{\text{GABA}}} + \sum_{i \in \text{Inh}} w_{ij}(t)\delta_i(t)$$

$$\text{with} \quad \frac{\mathrm{d}g_j^{\text{AMPA}}}{\mathrm{d}t} = -\frac{g_j^{\text{AMPA}}}{\tau_{\text{AMPA}}} + \sum_{i \in \text{Exc}} w_{ij}(t)\delta_i(t) \quad \text{and} \quad \frac{\mathrm{d}g_j^{\text{NMDA}}}{\mathrm{d}t} = \frac{g_j^{\text{AMPA}}(t) - g_j^{\text{NMDA}}}{\tau_{\text{NMDA}}}, \tag{3}$$

with $w_{ij}(t)$ the connection strength between neurons $i$ and $j$ (unitless), $\delta_k(t) = \sum \delta(t - t_k^*)$ the spike train of pre-synaptic neuron $k$, where $t_k^*$ denotes the spike times of neuron $k$, and $\delta$ the Dirac delta. All neurons received input from 5k Poisson neurons, with 5% random connectivity and constant rate $r_{\text{ext}}$, randomly sampled from the uniform distribution $\mathcal{U}(5, 15)$Hz in each simulation. The recurrent connectivity was instantiated with random sparse connectivity (10%).

## 2.2 Synaptic plasticity parameterizations

In order to explore a wide range of possible plasticity rules, we defined parameterized functional forms for these rules (i.e., search spaces) enabling us to flexibly generate various rules, by changing the parameters of the function. We considered two such parameterizations:

For **polynomial plasticity rules**, each recurrent synapse underwent spike-timing dependent plasticity (STDP) [21]. The weight from neuron $i$ to $j$ (of type X and Y; $(X, Y) \in \{E, I\}$) evolved such that

$$\frac{\mathrm{d}w_{ij}}{\mathrm{d}t} = \eta \left[ \delta_i(t)(\alpha_{\text{XY}} + \kappa_{\text{XY}} x_j(t)) + \delta_j(t)(\beta_{\text{XY}} + \gamma_{\text{XY}} x_i(t)) \right], \tag{4}$$

with $\eta$ a fixed learning rate. The variables $x_i(t)$ and $x_j(t)$ trace the pre- and post-synaptic spike trains

$$\frac{\mathrm{d}x_i}{\mathrm{d}t} = -\frac{x_i}{\tau_{\text{XY}}^{\text{pre}}} + \delta_i(t) \quad \text{and} \quad \frac{\mathrm{d}x_j}{\mathrm{d}t} = -\frac{x_j}{\tau_{\text{XY}}^{\text{post}}} + \delta_j(t), \tag{5}$$

where $\tau_{\text{XY}}^{\text{pre}}$ and $\tau_{\text{XY}}^{\text{post}}$ are the time constants of the traces associated with the pre- and postsynaptic neurons, respectively. Thus, the plasticity updates at each connection type depended on 6 parameters: $\theta_{\text{XY}} = [\alpha_{\text{XY}}, \beta_{\text{XY}}, \gamma_{\text{XY}}, \kappa_{\text{XY}}, \tau_{\text{XY}}^{\text{pre}}, \tau_{\text{XY}}^{\text{post}}]$, for a total of 24 parameters across all connection types (Fig. 2A).

For the **MLP plasticity rules**, only the recurrent EE and IE synapses underwent spike-triggered updates (Fig. 4A). Change of the weight from neuron $i$ to $j$ (of type X and E, $X \in (E, I)$) was computed by running forward multi-layer perceptrons (MLPs) that took as inputs several synaptic variables relevant to plasticity[1]:

$$\frac{\mathrm{d}w_{ij}}{\mathrm{d}t} = \eta \Bigg[ \delta_i(t) \text{MLP}_{\text{XE}}^{\text{pre}} \left( x_i^{(2)}, x_j^{(1)}, w_{ij}, \langle V_j \rangle, C_j^{\text{exc}}, C_j^{\text{inh}} \right) +$$
$$\delta_j(t) \text{MLP}_{\text{XE}}^{\text{post}} \left( x_i^{(1)}, x_j^{(2)}, w_{ij}, \langle V_j \rangle, C_j^{\text{exc}}, C_j^{\text{inh}} \right) \Bigg], \tag{6}$$

with $C_j^{\text{exc}} = \langle g_j^{\text{E}}(E_{\text{E}} - V_j) \rangle$ and $C_j^{\text{inh}} = \langle g_j^{\text{I}}(E_{\text{I}} - V_j) \rangle$ co-dependent terms representing the activity of neighboring synapses [10] (here all synapses with the same postsynaptic neuron), and $\langle V_j(t) \rangle$ the low-pass filtered membrane potential [11]. In this search space, time constants of synaptic traces were fixed at $\tau_{\text{EE}}^{(1)} = \tau_{\text{IE}}^{(1)} = 10$ms and $\tau_{\text{EE}}^{(2)} = \tau_{\text{IE}}^{(2)} = 100$ms [22, 7, 8, 10]. We chose a two-hidden-layer MLP, composed of 50 sigmoidal units in the first hidden layer, and 4 in the second. We trained only the final layer, keeping all other layers fixed. While this is standard practice with pre-trained ML

---

[1]Note that all input variables to the MLP are functions of $t$, which we elide to lighten notation.

models, we randomly initialised the parameters of the frozen layers, rather than pre-training them. We verified that when using random initializations for the first 2 hidden layers ($\sim \mathcal{U}(\frac{-1}{n_{\text{inp}}}, \frac{1}{n_{\text{inp}}})$, where $n_{\text{inp}}$ is the number of input features at a given layer), and only training the weights, bias, and output learning rate of the final layer, previously observed plasticity rules could be approximated (Supp. 6.7).

A given plasticity rule (EE or IE) thus consisted of one MLP (with 5 parameters: 4 weights and 1 bias) to update the synaptic weight when a pre-spike occurred, another MLP updating the synaptic weight for post-spikes (Fig. 4B), and a common learning rate $\eta_{\text{XE}}$ for both MLPs, resulting in 11 parameters per rule, and a total of 22 parameters between EE and IE.

## 2.3 Mean-Field predictions

In the polynomial case, we performed a self-consistent analysis of the weight and activity dynamics in the network at steady state, following previous work [23, 7, 10] (more details for all connection types in Supp. 6.3). For example, at EE synapses, we could calculate an expected firing rate $r_{\text{exc}}^*$ from the parameters of the rule such that:

$$r_{\text{exc}}^* = \frac{-\alpha_{\text{EE}} - \beta_{\text{EE}}}{\lambda_{\text{EE}}}, \tag{7}$$

with $\lambda_{\text{EE}} = \kappa_{\text{EE}} \tau_{\text{EE}}^{\text{post}} + \gamma_{\text{EE}} \tau_{\text{EE}}^{\text{pre}}$

## 2.4 Assessing network plausibility with metrics

To assess whether a particular plasticity rule was a suitable candidate, we simulated a spiking neuronal network (SNN) with that rule and quantified the resulting dynamics with multiple metrics on the activity and weight traces. The metric choices and values were informed by—and in line with—a range of modeling and experimental cortical studies [7, 8, 24–34]. For every simulation, we stored the raw spike trains and weight traces for 10s after 110s of simulated time (60s for the MLP rules). We grouped 15 individual metrics into four main criteria:

**Stable activity dynamics**: Rules were flagged as suitable when the excitatory and inhibitory population firing rates were bounded between 1 and 50 Hz, i.e., $r_{\text{exc}} \in [1, 50]$Hz and $r_{\text{inh}} \in [1, 50]$Hz, in line with a large swathe of literature reporting cortical firing rates of a few Hertz on average [24, 29, 30].

**Stable weight dynamics**: Fraction of synaptic weights that reached extreme values (0 or $w_{\text{max}}$), $f_{w_{\text{blow}}}$; mean absolute change in synaptic weights between simulation start and end across all synapse types, $w_{\text{creep}}$; mean weight at the last time-step across all synapses of a specific type, $\langle w_{\text{XY}} \rangle$. These metrics excluded rules whose weights changed too rapidly, did not converge, or reached unrealistic values. Rules were flagged as suitable if they satisfied $f_{w_{\text{blow}}} < 0.1$, $w_{\text{creep}} < 0.05$, $\langle w_{\text{EX}} \rangle < 0.5$ and $\langle w_{\text{IX}} \rangle < 5$ simultaneously.

**Near-irregular dynamics**: Mean coefficient of variation of the inter-spike intervals across all excitatory neurons, $\langle \text{cv}(\text{ISI}_i) \rangle$; auto-covariance for each spike-train, and its integral, averaged over time and the neural population, $\langle \rho \rangle_{i,t}$; Fano factor for each spike train averaged over the population, $\langle \text{Fano} \rangle_i$; standard deviation of the firing rate of individual neurons averaged across the population, $\langle \sigma \rangle_i$. These metrics quantified the regularity of a spike train, but, either due to metric assumptions or hyperparameter choices, each metric exhibited a different failure mode, which could be filtered out by combining the metrics. Rules were flagged as suitable if they satisfied $\langle \text{cv}(\text{ISI}_i) \rangle > 0.7$, $\langle \rho \rangle_{i,t} < 0.1$, $\langle \text{Fano} \rangle_i \in [0.5, 2.5]$ and $\langle \sigma \rangle_i < 5$ simultaneously.

**Near-asynchronous dynamics**: Standard deviation of the excitatory population firing rate, $\sigma_{r_{\text{exc}}}$; the Fano factor averaged across the population and then across time windows, $\langle \text{Fano} \rangle_t$; we also computed the population average of binned spike-trains, Fourier-transformed it and averaged the resulting power spectrum across all frequencies $f$ (except 0), $< S >_{i,f}$. Rules were flagged as suitable if they satisfied $\langle \text{Fano} \rangle_t \in [0.5, 2.5]$, $< S >_{i,f} < 1$ and $\sigma_{r_{\text{exc}}} < 0.05$ simultaneously. These bounds enforced the broadly asynchronous-irregular activity observed in cortex [24, 25, 27, 29, 30]. Ranges for metrics assessing synchrony/regularity were thus devised with independent Poisson spike trains in mind, with leeway for spatiotemporal correlations, as seen in cortical recordings [32–34].

Rules were considered plausible if they fulfilled all four broad criteria above (more details and analysis of individual metrics in Supp. 6.2).

## 2.5 Filter simulation-based inference

Having defined plasticity rules search spaces and network metrics, we turned to the problem of exploring these spaces, systematically filtering out unsuitable rules.

**Simulation-based inference (SBI)** is broadly defined as an approach for performing Bayesian inference given stochastic models (with tunable parameters $\theta$, and generating samples $x$) in cases where the likelihood of the model $p(x|\theta)$ is analytically inaccessible but simulation is possible [35–37]. SBI infers the posterior distribution $p(\theta|x)$, using samples from a prior distribution $\theta \sim \pi(\theta)$ and corresponding simulations $x \sim p(x|\theta)$, i.e., it applies Bayes' rule such that $p(\theta|x) \propto p(x|\theta)\,\pi(\theta)$, where $p(\theta|x)$ represents the distribution over model parameters most likely to have generated $x$.

This is an appealing framework for the problem of identifying plausible plasticity rules: we treated the parameters of the plasticity rule as parameters $\theta$ of a stochastic plastic spiking network that generated simulations $x$ consisting of the metrics defined in Section 2.4. We then used SBI—e.g., Neural Posterior Estimation (NPE) [35, 38, 39]—to identify the posterior distribution over plasticity rule parameters that were most likely to have resulted in plausible network dynamics. However, as we could not know beforehand which parameters were most likely to establish plausible dynamics, we had to define a broad prior over the parameters of the plasticity rule $\pi_0(\theta)$. In turn, the broadness of the prior meant that potentially millions of SNN simulations would be needed to find a few plausible rules, rendering this search method computationally unfeasible [40, 41].

In order to address this issue, we developed **filter SBI (fSBI)**. In fSBI, we applied the metrics quantifying plausibility *sequentially*, rather than all at once (Fig. 1B). Towards this goal, we first sampled plasticity rule parameters $\theta$ from a uniform prior and simulated an SNN with the corresponding rules. Next, we computed the various metrics defined in Section 2.4. We then used NPE to fit a posterior density over the plasticity rule parameters. Finally, we sampled all parameters that satisfied a given criterion from the posterior, e.g., stable network activity $p(\theta|r_{\text{exc}} \in [1,50]\text{Hz} \cap r_{\text{inh}} \in [1,50]\text{Hz})$, in the first instance, effectively "filtering out" other rules from the prior. These posterior samples were then used as the prior samples for the next round of inference, in which an additional criterion was applied to obtain a new, narrower posterior. Mathematically speaking, a filtering round $k$ using metrics $m_k = f_k(x)$ and corresponding conditions $g_k(m_k)$ proceeds as follows:

$$\pi_k(\theta) = p_{k-1}(\theta|m_{k-1} \ s.t. \ \mathbb{I}_{g_{k-1}}(m_{k-1})) \tag{8}$$

$$p_k(\theta|m_k) \propto p(m_k|\theta)\,\pi_k(\theta) \tag{9}$$

$$\pi_{k+1}(\theta) = p_k(\theta|m_k \ s.t. \ \mathbb{I}_{g_k}(m_k)), \tag{10}$$

where $\mathbb{I}$ is an indicator function over the condition $g$. This process is repeated until only plausible candidate rules remain, for a total of around 200k simulated rules per search space.

The distribution obtained at each filtering round is not necessarily a true Bayesian posterior, since (a) we change the metric on which we condition the posterior at each round, (b) we sample the posterior for a *range* of metric values rather than a single value, and (c) we also do not correct for the fact that at each round, we are sampling from the posterior of the previous round rather than from the prior [42] (more detailed analysis of this "pseudo"-posterior in Supp. 6.5).

## 3 Results

In our attempt to discover plasticity rules compatible with a given function, we introduced fSBI, a meta learning method that sequentially filters out unfit candidates rules from vast sets of possible plasticity rules in large recurrent spiking networks (SNNs, Methods. 2.1,10). We first applied fSBI in the polynomial search space described in Section 2.2, and compared the resulting distributions of rules to those predicted by mean-field analysis. We then applied fSBI to plasticity rules which themselves were parameterized by neural networks. In both cases, fSBI inferred expected and novel rules that robustly establish plausible dynamics in SNNs, while also eliminating some theoretically viable solutions.

## 3.1 fSBI infers plausible rules from a polynomial search space

We initially considered 4 co-active plasticity rules, E-to-E (EE), E-to-I (EI), I-to-E (IE) and I-to-I (II) in a recurrent spiking network of 5120 conductance-based leaky integrate-and-fire neurons (Fig. 2A). The network architecture and parameters were chosen following Zenke et al. [8]. Each

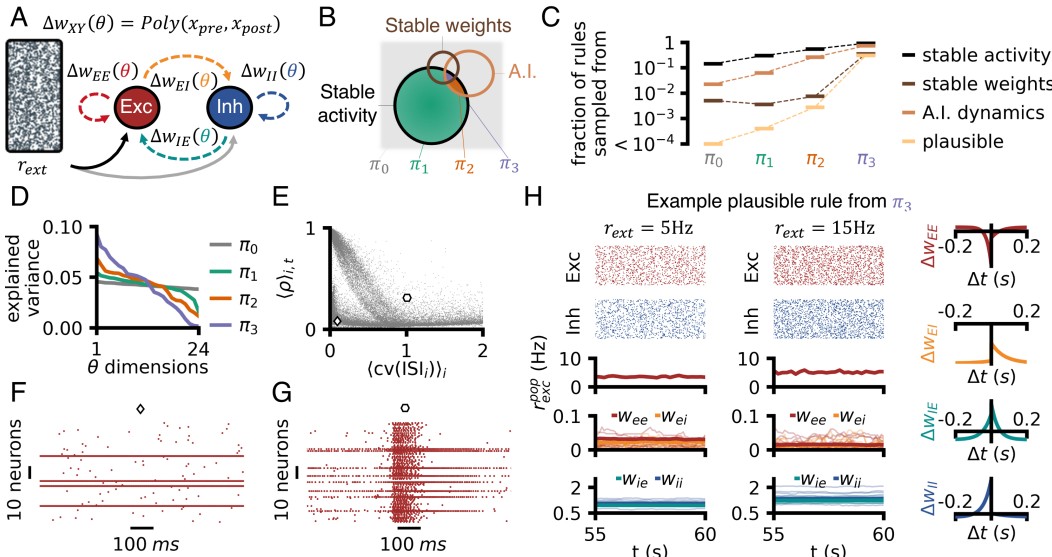

Figure 2: *Posterior of plausible plasticity rules with fSBI. Polynomial search-space.*
**A**: Recurrent spiking network receiving Poisson inputs at a random rate $r_{ext}$, with each recurrent connection parameterized with an STDP plasticity rule. Dotted lines denote plastic connections. **B**: Starting from uniform draws ($\pi_0$) from the polynomial search space, fSBI progressively filtered out the rules that do not establish plausible dynamics in an SNN ($\pi_1, \pi_2, \pi_3$), A.I. stands for asynchronous irregular. **C**: Fraction of rules sampled from each intermediate fSBI posterior that obey conditions on the corresponding network metrics. **D**: Principal component analysis (PCA) on samples from each intermediate fSBI posterior ($\pi_0 \rightarrow \pi_3$). Variance explained along the corresponding PCA directions across rounds. **E**: For each plastic network simulated in fSBI (pooled across all rounds), the corresponding CV and auto-covariance (Methods. 2.4). **F**: Example edge-case of the auto-covariance metric: network fulfills the auto-covariance condition ($\rho < 0.1$), but not the CV condition ($\langle cv(ISI_i)\rangle \neq 1$). **G**: Example edge-case of the CV metric: network fulfills the CV condition, but not the auto-covariance condition (Fig. 2G). **H**: Left: Example plausible candidate rule simulated in two networks with different input rates. Right: pre-post pairing protocol on the 4 co-active rules.

rule was parameterized as a polynomial (Eqn 6, [15]), with a total of 24 parameters $\theta$ (6 parameters per connection type). We applied fSBI to this 24-dimensional search space, to identify and retain the rules that established biologically plausible dynamics in SNNs. To ensure the inferred rules robustly enforced plausible dynamics in SNNs, any simulation was assigned a random input rate and connectivity matrix.

We started fSBI with uniform priors for all plasticity parameters: $\tau_{XY}^{pre}, \tau_{XY}^{pre} \sim \mathcal{U}[10, 100]$ms, and $\alpha_{XY}, \beta_{XY}, \gamma_{XY}, \kappa_{XY} \sim \mathcal{U}[-2, 2]$. The resulting rules formed the initial prior $\pi_0$. We then sampled multiple plasticity rules $\theta \sim \pi_0$, simulated SNNs with these plasticity rules, and saved their spike-trains and weight evolutions for offline analysis of the network metrics $x$ (see Section 2.4). For the first round, we trained Neural Posterior Estimation (NPE) to produce posteriors conditioned only on a subset of metrics—the excitatory and inhibitory population firing rates—$p(\theta|r_{exc}, r_{inh})$. Next, we sampled plasticity rules from this posterior corresponding to networks with excitatory and inhibitory rates between 1 and 50 Hz, i.e., $\theta' \sim p(\theta'|r_{exc} \in [1, 50]$Hz$, r_{inh} \in [1, 50]$Hz$)$. These rules were used for the next round of inference, which we conditioned on the metrics associated with stable weight dynamics ($f_{w_{blow}}, w_{creep}, \langle w_{XY}\rangle$), in addition to the population activity rates. We thus made the posterior from the previous iteration, $p(\theta|r_{exc} \in [1, 50]$Hz$, r_{inh} \in [1, 50]$Hz$)$, into the prior $\pi_1(\theta)$ of the current iteration. We progressively applied all above mentioned metrics (defined in Section 2.4), in the order of the least to the most restrictive metric, to maximize sample efficiency (Fig. 2B, C).

We quantified the behaviour of the network with a range of metrics [23, 25] and noticed that some networks satisfied some conditions, but not others which should discern similar qualities. For example, some networks passed the auto-covariance criterion ($\langle\rho\rangle_{i,t} < 0.1$) but not the CV criterion ($\langle cv(ISI_i)\rangle > 0.7$), by allowing a small population of neurons with unrealistically high firing rates to "cheat" the metric (Fig. 2E, F). Conversely, other networks passed the CV condition but failed the auto-covariance one, e.g., with a bimodal inter-spike interval (ISI) distribution (Fig. 2G). To

filter out these edge cases, we used several metrics for each desired aspect of the network dynamics (Section 2.4).

We found that fSBI successfully identified relevant parts of the search space: the 10,000 samples from the uniform prior $\pi_0(\theta)$ contained no plausible rules (i.e., no rules producing networks satisfying all criteria). However, by sequentially adding constraints we arrived at a posterior $\pi_3(\theta)$ that, at the end of the fourth search round, led to more than 50% suitable rules (Fig. 2C). Interestingly, the dimensionality of the posterior, i.e., of the "suitable rule space", shrunk as we sequentially refined the criteria for plausible network dynamics (Fig. 2D), suggesting that fSBI captured specific relationships between the plasticity parameters. On the other hand, the final posterior remained relatively high dimensional, comprising a variety of different rules, with no apparent posterior collapse onto a single rule. The plausible rules sampled from this final posterior showed stable activity and weight dynamics, regardless of their input rates (Fig. 2H).

**Interdependencies between plasticity parameters in the fSBI inferred rules**: Next, we examined interdependencies between fSBI-inferred parameters as a possible explanation for the reduced effective dimensionality of the parameter space (Fig. 2D). We found that the correlation was overall weak between the parameters of the plausible rules (Supp. Fig. 6.2B). This was not surprising, as we compared a range of rules that establish a wide diversity of SNN dynamics, such that individual rules can be expected to have different parameter inter-dependencies. We thus narrowed our investigation to rules that produced excitatory firing rates of 10Hz (as opposed to a range of rates between 1 and 50Hz, Fig. 3A). The corresponding correlation matrix revealed a fine-grained structure for parameters within the same rule (block diagonal structure), where we observed mostly negative correlations, especially between the non-Hebbian parameters within each rule, $\alpha_{XY}$ and $\beta_{XY}$. Additionally, we found both positive and negative correlations between any given connection types (off-diagonal structure, Fig. 3A), suggesting that plasticity mechanisms must be co-tuned to function properly.

**fSBI rules improve upon mean-field predictions**: We also compared the fSBI-inferred rules to mean-field predictions. Mean-field analysis links the plasticity rule parameters to the average expected firing rate (Section 2.3, Fig. 3B), thus allowing us to obtain theoretically viable rules. Samples from the initial prior did not show any similarity to theoretical predictions when projected along the mean-field axes (Fig. 3C, left). As rules were filtered by progressively more constraints, a similar structure to the mean-field predictions emerged, albeit confined to firing rates below 10Hz, possibly because of synchronisation effects for higher firing rates (Fig. 3C for EE; similar conclusions for the other three synapse types in Supp. Fig.6.2). Interestingly, the set of rules predicted by mean field theory was not a superset of those selected by fSBI: Some rules were exclusively predicted by one but not the other approach (Fig. 3D, E). Notably, many rules predicted by mean-field, but *not* by fSBI, did not result in plausible dynamics when tested numerically (Fig. 3D, middle). In contrast, several rules predicted by fSBI but rejected by mean-field proved numerically viable (Fig. 3E, middle). Finally, the EE rules predicted by both mean-field and fSBI were plausible (Fig. 3D, bottom). We noted that fSBI rules lead to plausible network dynamics both when simulated with only the EE parameters (keeping the weights of the other connection types fixed, Fig. 3E, bottom), and also when using the rule for all connection types (Fig. 3E, middle), thus confirming the success of fSBI.

**Comparing rules under pre-post protocols**: We visualized the fSBI rules under the pre-post pairing protocol widely used experimentally and theoretically to uniquely identify STDP rules [1, 6, 43, 44]. We found that among the various rules simulated, there was no apparent link between how rules appeared in the pre-post protocol, and their ability to establish plausible dynamics in SNNs. Some rules appeared near-identical under the pre-post visualization and yet resulted in vastly different network dynamics (Fig. 3D). Conversely, rules with visibly differing pre-post protocols established similar network dynamics (Fig. 2H and Fig. 3E). Taken together, these results question the relevance of such classical pre-post protocols in elucidating and uniquely determining the plasticity rules at play in the brain, particularly when they are performed without additional disambiguating experiments.

### 3.2  fSBI infers plausible rules from an MLP search space

We wanted to expand our search space to include local and semi-local variables that have been theoretically or experimentally linked to synaptic changes [2–4], such as the efficacy of the synapse [44, 8], the post-synaptic membrane potential [11], the spike-history of both the pre- and post-synaptic neurons [22], as well as the activity of neighboring excitatory and inhibitory synapses (co-dependent plasticity [10]). Since flexibly combining these variables with a polynomial would result in very

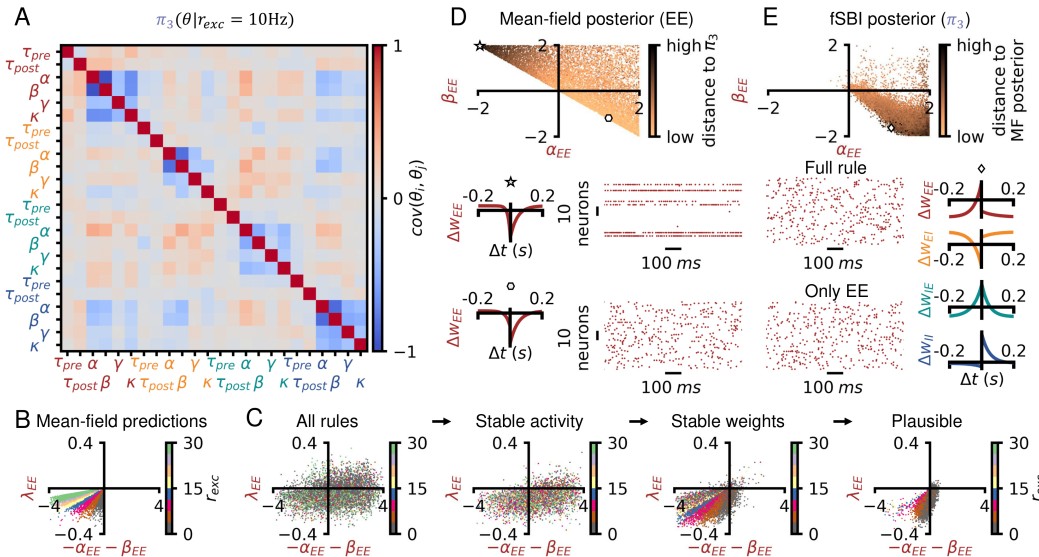

Figure 3: *Polynomial fSBI rules compared to rules predicted by mean-field.*
**A**: Pearson correlation matrix of plasticity rules sampled from the final fSBI posterior ($\pi_3$), conditioned on the excitatory firing rate $r_{\text{exc}} = 10$Hz. **B**: EE rules predicted by mean-field, plotted alongside the relevant parameter combinations (eq. 7). **C**: Rules sampled from fSBI posteriors, with an increasing number of conditions, along the same axes as in B. **D**: Same set of rules as in B, but shown along 2 of the 6 plasticity parameters ($\alpha$ and $\beta$), color coded by the mean Euclidean distance between each rule and the 10 closest fSBI-inferred EE rules. Middle (resp. bottom), network dynamics with a rule among the furthest (resp. closest) from fSBI samples, and the corresponding pre-post protocol. **E**: Rules sampled from the final fSBI posterior $\pi_3$, shown along the same axes as D, color coded by the mean Euclidean distance between each rule and the 10 closest mean-field plasticity rules. Below, network dynamics with one of the rules furthest from mean-field, and the corresponding pre-post protocol, either the full rule (EE, EI, IE, II, middle) or only the EE part of the rule (bottom). Weight traces in Supp. 6.6.

high-dimensional, hard-to-navigate search spaces, we switched to multi-layer perceptrons (MLP). The MLP-based rules we designed used the aforementioned synaptic variables as inputs at the time of a spike, and produced a weight change for a given synapse (Fig. 4B), thus representing the plasticity rule. Crucially, we only trained the final layer of that MLP, thus allowing for potentially highly non-linear interactions between the plasticity variables while retaining a relatively low dimensional (searchable) parameter space. We verified that such MLP-parameterized rules can approximate established plasticity rules on supervised tasks (see Supp. Fig.6.6), and then applied fSBI to inferring co-active EE and IE rules (Fig. 4A, B). Like for the polynomial space, we found that fSBI produced progressively more constrained posteriors, such that the final fSBI posterior generated over 50% plausible rules (Fig. 4C). Furthermore, fSBI reduced the effective dimensionality of the parameter space, i.e., increased correlations between the parameters (Supp.6.7), like what we had observed for the polynomial expression of rules. Interestingly, plausible rules from the final fSBI posterior produce stable activity and weight dynamics, but with unique pre-post protocol profiles that are markedly different from that of experimentally observed rules, as well as the rules discovered by the polynomial method above (Fig. 4D, E, Supp.6.7). These rules represent potentially complex and non-linear interactions between Hebbian, local and semi-local factors, but nevertheless robustly establish plausible network dynamics.

**Generalization capabilities of meta-learned rules**: Finally, we verified that both the polynomial and the MLP rules obtained with fSBI generalized to different network sizes, sparsity, weight initializations, and ratios of E to I, although a few metrics were sometimes outside the predefined ranges (Fig. 5A). We then tested the "plausible" rules on 20-minute-long simulations (versus 2min in Fig. 2&4). Approx. 25% of the polynomial rules still met the conditions on all 15 metrics after 20 minutes (Fig. 5B). Most other rules were disqualified by our $\langle w_{\text{IE}} \rangle$ and $\langle w_{\text{II}} \rangle$ conditions, which select long-term stable weights, and are thus sensitive to simulation duration. Correspondingly, the II and IE plasticity parameters appeared more refined compared to the starting set (Fig. 5D). Overall, this

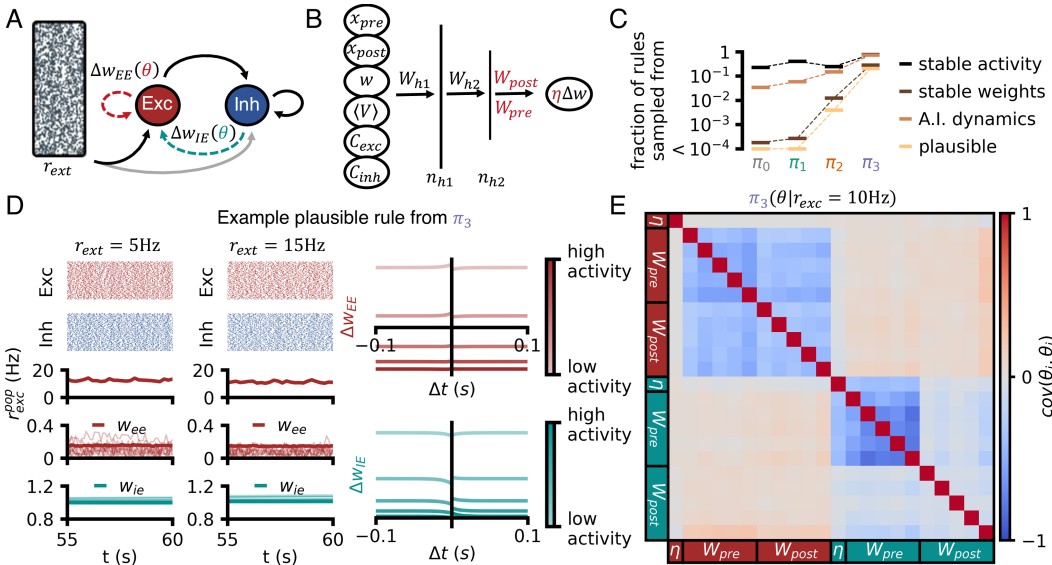

Figure 4: *Posterior of plausible plasticity rules with fSBI. MLP search-space.*
**A**: Recurrent spiking network receiving Poisson inputs at a random rate $r_{ext}$, with recurrent EE and IE plastic connections parameterized as multi-layer-perceptrons (MLP). **B**: Weight changes via a forward pass through an MLP that depends on pre- and post-synaptic traces, current synaptic weight, smoothed membrane potential and two codependency terms at every spike (Methods. 2.2). **C**: Fraction of rules sampled from each intermediate fSBI posterior that obey the corresponding network metric condition. **D**: Left: Example plausible candidate rule simulated in two networks with different input rates. Right: pre-post pairing protocol on the EE and IE rules. Note that these rules have different outcomes on the pre-post protocol depending on the network activity (Supp.6.4). **E**: Pearson correlation matrix of the plasticity rules sampled from the final fSBI posterior, conditioned on the excitatory firing rate being 10Hz.

suggested that the 2min task had properly constrained the EE and EI rules, but was too short to fully constrain the II and IE rules. Similarly, 35% of the 2min-plausible MLP rules met all conditions in the 20min task. Interestingly, dimensionality reduction of the successful plasticity parameters did not reveal sub-structures, suggesting that the 20min task did not dramatically refine the set of filtered rules (Fig. 5C).

# 4 Discussion and Outlook

Plasticity mechanisms are arduous to probe *in vivo* and theoretically challenging to describe due to the complex interplay between neural activity and weight dynamics. Here, we introduce fSBI, a tool to numerically infer families of plasticity rules while enforcing strong plausibility constraints on SNN dynamics. We show that fSBI can discover generally plausible plasticity rules in large SNNs, with rules parameterized either as polynomials or MLPs.

fSBI has several advantages compared to previous approaches: analytical methods require various and sometimes strong assumptions; local optimization methods only provide single solutions; and brute-force approaches do not scale (in this study, randomly drawing from the full space of plasticity rules would have required simulating ∼100M rules, compared to 200k, Fig. 2C, 4C). However, these approaches are not mutually exclusive: analytical frameworks allow us to understand the sets of rules inferred by fSBI and produce meaningful experimental predictions. Furthermore, the fSBI distributions of plausible rules are a promising starting point for further investigation into specific network computations, potentially suited to local optimization approaches.

In contrast with previous *in silico* studies of plasticity that produce single predictions, fSBI is a systematic approach to make predictions based on many (if not all) candidate plasticity rules within a search space. For instance, we showed how an isolated EE rule may suggest a Hebbian mechanism, but the fSBI-discovered set contains equally valid anti-Hebbian rules that can establish near identical

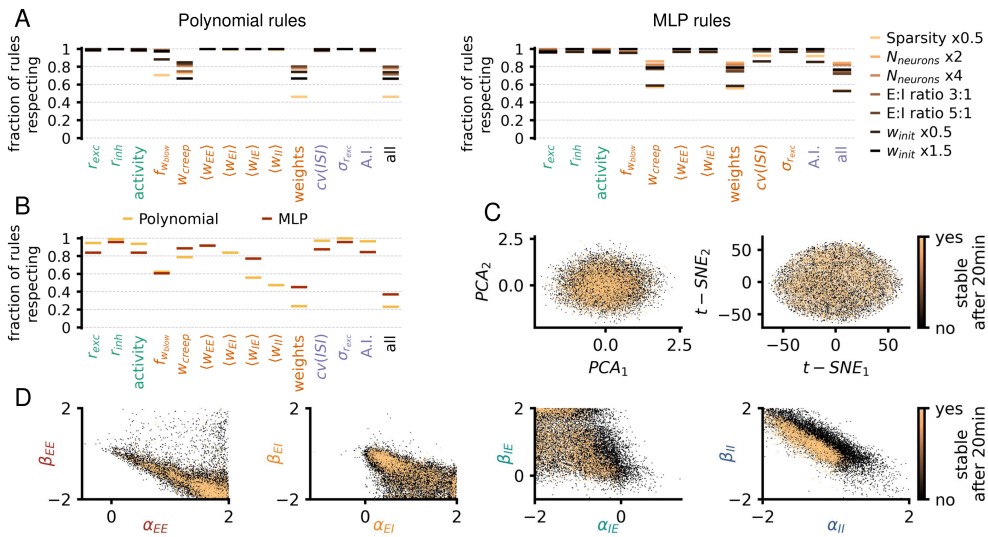

Figure 5: *Robustness of polynomial and MLP rules.*
**A**: 1k polynomial (left) and 1k MLP (right) rules that fulfilled all plausibility conditions on the 2 min task (Fig.2&4) were simulated on the same task, but with a different probability of connection (5%), network size (10k, 20k), E:I ratios or initial weights ($w_{\text{EE}}^{\text{init}} = w_{\text{EI}}^{\text{init}} = \frac{w_{\text{IE}}^{\text{init}}}{10} = \frac{w_{\text{II}}^{\text{init}}}{10} = 0.05$, or $w_{\text{EE}}^{\text{init}} = w_{\text{EI}}^{\text{init}} = \frac{w_{\text{IE}}^{\text{init}}}{10} = \frac{w_{\text{II}}^{\text{init}}}{10} = 0.15$). All simulations had a random input strength and initial connectivity. "Activity", "weights" and "A.I" denote the intersection of all metrics in that category. **B**: 10k "plausible" polynomial and 10k MLP rules were simulated on a 20-minute extended version of the 2-minute background task (Fig.2&4). Fraction of the rules that fulfill some plausibility conditions on the 20-minute task. **C**: First two features of PCA (left), t-SNE (right) applied to the MLP rules in B. Rules fulfilling all plausibility conditions on the 20-minute task are in yellow. **D**: Non-Hebbian parameters of the polynomial rules from B, with the same color code as C.

network activity (Figs. 2H, 3E), suggesting that studies of single rules or individual connection types may be grossly under-constrained.

More generally, our study challenges the widely used pre-post pairing protocols [1, 6, 44], when used in isolation. Indeed, rules that look near identical under such pre-post protocols may have widely different network-level effects (Fig. 3D, middle and bottom), validating, in hindsight, the need for finely-tuned orchestration in previous studies [8]. Conversely, rules that appear different under observation with a pre-post protocol (Fig. 2H and Fig. 3E) may lead to similar network dynamics. In addition, the results above were obtained in noise-free *in silico* experiments, suggesting that the connection between empirical data and plasticity rules may be even more complex. The interpretability of the pre-post pairing protocols is further compromised in the MLP-based rules, since these rules include factors linked to plasticity that are not explicitly constrained by such low-dimensional protocols, leading to widely different outcomes depending on the synaptic and neuronal state (Fig. 4D). Additionally, we have not touched on the effects of neuromodulation in our study, something that is currently out of scope but will become the focus of future studies.

Our approach naturally comes with limitations and idiosyncrasies. First, we cannot prove the long-term stability of the rules, as we are limited by the numerical approach and associated demanding compute requirements. On longer timescales, slower homeostatic mechanisms may play a part in synaptic dynamics [45, 46]. Another potential limitation of fSBI is the curse of dimensionality, inherent to most SBI approaches [36], although some recent improvements have been proposed [47–50]. Finally, further analysis is needed to forge robust and meaningful predictions from the commonalities of the large set of inferred rules.

In summary, we meta-learn plasticity rules with an inference-based method, fSBI. This method can be applied to large spiking networks with flexibly parameterized plasticity rules and narrow down the set of promising plasticity mechanisms.

## 5    Acknowledgments

We thank Chaitanya Chintaluri, Everton Agnes, Nicoleta Condruz, Douglas Feitosa Tomé, Michael Deistler and Jan Boelts for helpful discussions and feedback on the manuscript. This work was funded by the European Research Council (ERC consolidator grant SYNAPSEEK), the German Research Foundation (DFG; Germany's Excellence Strategy MLCoE – EXC number 2064/1 PN 390727645), the German Federal Ministry of Education and Research (BMBF; Tübingen AI Center, FKZ: 01IS18039A), the Human Frontier in Science Program (RGY0076/2018) and the FENS-Kavli Network of Excellence scientific exchange program. This research was supported by the Scientific Service Units (SSU) of IST Austria through resources provided by Scientific Computing (SciComp).

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
