# 6 Supplementary material

All spiking network simulations were run using Auryn [51] on single cpu-cores. Network simulations performed in fSBI were distributed over 300 cores from four workstations for two months. Simulations were analyzed using numpy, scipy, sklearn and pytorch [52–55]. Inference was performed using the sbi package [56]. Figures were made using matplotlib [57]. Code for simulations, analysis and figures is available on https://github.com/VogelsLab/fSBI.

## 6.1 Spiking network simulations

Table 1:

| Neuron model parameters (E and I populations) | | |
| --- | --- | --- |
| Membrane time-constant | $\tau_m$ | 20ms |
| Resting potential | $V_{\text{rest}}$ | $-70$mV |
| Excitatory reversal potential | $E_{\text{E}}$ | 0mV |
| Inhibitory reversal potential | $E_{\text{I}}$ | $-80$mV |
| AMPA/NMDA ratio | $a$ | 0.3 |
| GABA time-constant | $\tau_{\text{GABA}}$ | 10ms |
| AMPA time-constant | $\tau_{\text{AMPA}}$ | 5ms |
| NMDA time-constant | $\tau_{\text{NMDA}}$ | 100ms |
| Threshold resting state | $V_{\text{base}}^{\text{th}}$ | $-50$mV |
| Threshold time-constant | $\tau_{\text{th}}$ | 5ms |
| Threshold increment on spike | $V_{\text{spike}}^{\text{th}}$ | 100mV |

Table 2:

| Network parameters | | |
| --- | --- | --- |
| Number of excitatory neurons | $N_{\text{E}}$ | 4096 |
| Number of inhibitory neurons | $N_{\text{I}}$ | 1024 |
| Initial EE weights | $w_{\text{EE}}$ | 0.1 |
| Initial EI weights | $w_{\text{EI}}$ | 0.1 |
| Initial IE weights | $w_{\text{IE}}$ | 1 |
| Initial II weights | $w_{\text{II}}$ | 1 |
| Sparsity (EE, EI, IE, II) | $\xi_{\text{rec}}$ | 0.1 |
| Number of input neurons | $N_{\text{ext}}$ | 5000 |
| Sparsity inputs (Ext-E, Ext-I) | $\xi_{\text{ext}}$ | 0.05 |
| Input weights (Ext-E, Ext-I) | $w_{\text{ext}}$ | 0.075 |

Table 3:

| Simulation parameters | | |
| --- | --- | --- |
| Simulation time polynomial rules (no recording) | $t_{\text{init}}^{\text{poly}}$ | 120s |
| Simulation time MLP rules (no recording) | $t_{\text{init}}^{\text{MLP}}$ | 60s |
| Recording time | $t_{\text{rec}}$ | 10s |
| Simulation time-step | $\Delta t_{\text{auryn}}$ | 0.1ms |

**Early stopping of simulations**: To save compute-time during simulation of plastic SNNs, specifically in the earlier rounds of fSBI, we used an online estimate of the network population activity (exponential kernel with time constant 1s) and stopped the simulation if the network displayed a population activity above 100Hz.

Note that in the networks with an MLP rule, adaptation (spike-triggered self-inhibition, with time-constant $\tau_{\text{sfa}} = 100$ms) was added to both neuron populations.

## 6.2 Network metrics

Given spike-trains $(S_i(t))_{i \in E}$ and weight evolutions $(w_{ij}(t))_{i,j \in E,I}$ during a recording window $[t_{\text{start}}, t_{\text{stop}}]$, and consistent with previous studies [23, 25], we introduced multiple network metrics, defined as functions taking spike-trains or weight traces as inputs and outputting scalars: $M : \left\{ S_i(t), w_{ij}(t) \right\}_{i,j \in E} \to \mathbb{R}$. Below we detail the metrics chosen in this study. In practice, the activity of 1000 random excitatory neurons, 500 inhibitory neurons, and 100 weights for each plastic connection type was recorded and stored for each network simulation.

### 6.2.1 Stable activity

**Excitatory/inhibitory population firing rate** $r_{\text{exc}}$ and $r_{\text{inh}}$: number of spikes produced by all neurons during the recording window, divided by number of neurons and recording length. Rules were flagged as suitable if $r_{\text{exc}} \in [1, 50]$Hz and $r_{\text{inh}} \in [1, 50]$Hz. The lower bound of 1Hz was chosen mainly for numerical reasons: allowing us to collect enough spikes in a given 2min simulation and thus meaningfully assess rules with spike-triggered updates.

### 6.2.2 Stable weights

**Weigth blow-up** $f_{w_{\text{blow}}}$: the fraction of blown-up weights ($w = 0$ or $w = w_{\text{max}}$) across the recording was computed and averaged across all plastic connection types. The condition chosen was $f_{w_{\text{blow}}} < 0.1$. This metric was introduced to reject biologically unrealistic rules that led more than 10% of the weights of any synapse type to 0 or to the maximum weight .

**Weight creep** $w_{\text{creep}}$: the maximum change in relative mean weights across all plastic connection types across the recording window: $w_{\text{creep}} = \max \left( 2 \frac{|\langle w_{ij}^{\text{XY}} \rangle_{t_{\text{stop}}} - \langle w_{ij}^{\text{XY}} \rangle_{t_{\text{start}}}|}{\langle w_{ij}^{\text{XY}} \rangle_{t_{\text{stop}}} + \langle w_{ij}^{\text{XY}} \rangle_{t_{\text{start}}}} \right)_{\text{XY}}$. The condition chosen was $w_{\text{creep}} < 0.05$. This metric was designed to reject weights that changed unrealistically fast, excluding mean relative changes of more than 5% over the final 10s of simulation.

**Mean final weight** $\langle w_{\text{XY}} \rangle$: The mean value of weights of each connection type at $t_{\text{stop}}$ was computed. The condition chosen was $\langle w_{\text{EY}} \rangle < 0.5$ (excitatory connections) and $\langle w_{\text{IY}} \rangle < 5$ (inhibitory connections). These ranges were chosen so that PSPs were smaller than 3 millivolts on average, in line with experimental data [29, 31].

### 6.2.3 Near-asynchronous activity

Several metrics were used to enforce the broadly asynchronous-irregular activity observed in cortex [24, 25, 27, 29, 30]. Ranges for these were devised with independent Poisson spike trains in mind, with leeway for spatiotemporal correlations, as seen in cortical recordings [32–34].

**Standard deviation of the population firing rate** $\langle \sigma \rangle_i$: the excitatory population firing rate was computed over successive 1ms time windows, on which the standard deviation was computed. The condition was $\langle \sigma \rangle_i < 5$. This measure has been widely used to detect network-wide events —synchrony—, corresponding to high standard deviations [23, 25].

**Fourier transform** $< S >_{i,f}$: spike trains pooled across all recorded neurons were binned over successive 1ms time windows. The resulting 1D signal was Fourier-transformed. The sum of the absolute value of the signal in Fourier space was computed, excluding the contribution of the mean firing rate (power at frequency $f = 0$Hz in Fourier space), and used as a metric with the condition $< S >_{i,f} < 1$.

**Fano factor (spatial)** $\langle \text{Fano} \rangle_i$: after binning the spike-trains over 100ms successive windows, the Fano factor across neurons was computed for each time window, then averaged over the windows. The condition used was $\langle \text{Fano} \rangle_{i,t} \in [0.5, 2.5]$.

### 6.2.4 Near-irregular activity

**CV** $\langle \text{cv}(\text{ISI}_i) \rangle_i$: the coefficient of variation of the inter-spike intervals (ISIs) for each neuron in the network was computed, then averaged over neurons: $\langle \text{cv}(\text{ISI}_i) \rangle_i = < \frac{\sigma_{\text{ISI}_i}}{\mu_{\text{ISI}_i}} >_i$. This measure has no hyper-parameters, and has been widely used in neuroscience for decades to assess the regularity of a spike-train [23, 25].

**Auto-covariance** $\langle\rho\rangle_{i,t}$: After binning the spike-trains with a window of 10ms, we computed the normalized, absolute value of the auto-covariance for each neuron between -500ms and 500ms. We then took the mean area under the curve and averaged over neurons. The condition was $\langle\rho\rangle_{i,t} < 0.1$. This metric also targets the regularity of spike-trains, but does not assume unimodality of ISIs like the cv.

**Fano-factor (temporal)** $\langle\text{Fano}\rangle_t$: after binning the spike-trains over 100ms successive windows, the Fano factor was computed per neuron, then averaged over windows. The condition was $\langle\text{Fano}\rangle_t \in [0.5, 2.5]$.

We chose the ranges for the less widespread metrics by benchmarking the respective metrics on example networks such as Poisson spike trains at various rates, networks from classical studies in the field [7–9] and also implausible networks e.g., synchronous-regular regime.

The choices of metrics and ranges made in this study are not universal, and a focus on specific brain regions, especially subcortical ones, may necessitate some changes. fSBI allows for such changes of metrics and corresponding ranges post hoc, at little computational cost, because the raw spike trains and weight traces are stored.

## 6.3   Mean-field analysis

In the polynomial search space, we performed a self-consistent analysis of the weight and activity dynamics in the network at steady state, as was done previously [23, 7, 10]. Such analysis is performed independently for every synapse-type and links the plasticity parameters and the population activity. If we consider the EE rule:

$$\frac{\mathrm{d}w_{ij}}{\mathrm{d}t} = \eta\left[\delta_i(t)(\alpha_{\text{EE}} + \kappa_{\text{EE}}x_j(t)) + \delta_j(t)(\beta_{\text{EE}} + \gamma_{\text{EE}}x_i(t))\right], \tag{6.1}$$

with $\delta_k(t) = \sum \delta(t - t_k^*)$ the spike train of neuron $k$, $t_k^*$ denotes the spike times of neuron $k$, and $\delta$ the Dirac delta. The variables $x_i(t)$ and $x_j(t)$ trace the pre- and post-synaptic spike trains:

$$\frac{\mathrm{d}x_i}{\mathrm{d}t} = -\frac{x_i}{\tau_{\text{EE}}^{\text{pre}}} + \delta_i(t) \quad \text{and} \quad \frac{\mathrm{d}x_j}{\mathrm{d}t} = -\frac{x_j}{\tau_{\text{EE}}^{\text{post}}} + \delta_j(t), \tag{6.2}$$

where $\tau_{\text{EE}}^{\text{pre}}$ and $\tau_{\text{EE}}^{\text{post}}$ are the time constants of the traces associated with the pre- and postsynaptic neurons, respectively. We can average this equation over a large time-window compared to the timescales involved:

$$\left\langle\frac{\mathrm{d}w_{ij}}{\mathrm{d}t}\right\rangle = \eta\left[\alpha_{\text{EE}}\langle\delta_i(t)\rangle + \kappa_{\text{EE}}\langle\delta_i(t)x_j(t)\rangle + \beta_{\text{EE}}\langle\delta_j(t)\rangle + \gamma_{\text{EE}}\langle\delta_j(t)x_i(t)\rangle\right], \tag{6.3}$$

Assuming the existence of steady-state on the weights and activities, and a homogeneous network of independent Poisson spike-trains, this steady state is given by:

$$\left\langle\frac{\mathrm{d}w_{ij}}{\mathrm{d}t}\right\rangle = 0, \quad \langle\delta_i(t)\rangle = \langle\delta_j(t)\rangle = r_{\text{exc}}^*, \quad \langle x_i(t)\rangle = \tau_{\text{EE}}^{\text{pre}}r_{\text{exc}}^*, \quad \langle x_j(t)\rangle = \tau_{\text{EE}}^{\text{post}}r_{\text{exc}}^*, \tag{6.4}$$

which leads to the equation used in Methods. 7. Similar analysis on the other connection types leads to:

$$\begin{array}{llll} \text{(EE)} & r_{\text{exc}}^* = \dfrac{-\alpha_{\text{EE}} - \beta_{\text{EE}}}{\lambda_{\text{EE}}}; & \text{(EI)} & r_{\text{inh}}^* = \dfrac{-\alpha_{\text{EI}}r_{\text{exc}}^*}{\beta_{\text{EI}} + \lambda_{\text{EI}}r_{\text{exc}}^*} \\[3mm] \text{(IE)} & r_{\text{exc}}^* = \dfrac{-\alpha_{\text{IE}}r_{\text{inh}}^*}{\beta_{\text{IE}} + \lambda_{\text{IE}}r_{\text{inh}}^*}; & \text{(II)} & r_{\text{inh}}^* = \dfrac{-\alpha_{\text{II}} - \beta_{\text{II}}}{\lambda_{\text{II}}}, \end{array} \tag{6.5}$$

with the notation $\lambda_{\text{XY}} = \kappa_{\text{XY}}\tau_{\text{XY}}^{\text{post}} + \gamma_{\text{XY}}\tau_{\text{XY}}^{\text{pre}}$.

In addition, the fixed-point should be stable, thus for EE:

$$\frac{\partial}{\partial r_{\text{exc}}}\left(\frac{\mathrm{d}r_{\text{exc}}}{\mathrm{d}t}\right)(r_{\text{exc}}^*) < 0. \tag{6.6}$$

We can then assume for an excitatory synapse (considering a linear rate model) that:

$$\frac{\mathrm{d}r_{\text{exc}}}{\mathrm{d}t} \propto \left\langle\frac{\mathrm{d}w}{\mathrm{d}t}\right\rangle, \tag{6.7}$$

which leads to $\alpha_{EE} + \beta_{EE} > 0$ and $\lambda_{EE} < 0$, and similar conditions on other synapse types.

Overall, such analysis provided a rough first guess of analytically plausible rules, against which to fit the rules recovered from fSBI. Further constraining across synapse-types could be done by adding that since we consider a recurrent network, $r_{\text{exc}}$ and $r_{\text{inh}}$ are related.

## 6.4  Pre-post pairing protocol

The pre-post pairing protocol was implemented by computing the total weight change of a synapse receiving one pre-synaptic spike and one post-synaptic spike with time lag $\Delta t = t_{\text{post}} - t_{\text{pre}}$.

For the case of an MLP rule, for which the other input variables were not constrained by the protocol, the protocol was repeated 5 times with several values for the other synaptic variables reflecting various overall levels of activity: from low to high. For an EE rule:

$$w_{ij} = 0.01 \to 1, \ \langle V_j \rangle = -70\text{mV} \to -50\text{mV}, \ C_j^{\text{exc}} = 0 \to 30\text{mV}, \ C_j^{\text{inh}} = 0 \to 30\text{mV}, \quad (6.8)$$

For an IE rule:

$$w_{ij} = 1 \to 10, \ \langle V_j \rangle) = -70\text{mV} \to -50\text{mV}, \ C_j^{\text{exc}} = 0 \to 30\text{mV}, \ C_j^{\text{inh}} = 0 \to 30\text{mV} \quad (6.9)$$

## 6.5  fSBI and true Bayesian posteriors

With fSBI, the final distribution we obtain after $T$ rounds of filtering *is not* the true Bayesian posterior $p(\theta | m_1, m_2, \ldots, m_T)$, since the metrics $m_1, \ldots, m_T$ we condition the posterior on change at each filtering round. We also do not correct for the change in prior at each round. Furthermore, when we replace the prior in each subsequent round with the posterior from the preceding round, we sample from it for a range of metric values, rather than a single value.

We here clarify the relationship between the pseudo-posterior obtained from fSBI and the corresponding true Bayesian posterior. We note that targeting the true Bayesian posterior (and the additional theoretical and computational burden it would add to the algorithm) is unnecessary, since our main objective is rather to sample plausible rules. Nevertheless, clarifying the relationship between the pseudo- and true posterior can help choose strategies in applying fSBI to recover plausible rules, as we describe below.

We infer fSBI posteriors in two ways: by conditioning *only* on new metrics at each filtering round, or conditioning on new metrics in addition to the metrics from previous rounds. We address the two cases separately and restrict our discussion to two filtering rounds, for clarity.

**New metrics only at each round**: We recall from Eqns. 10 that after two rounds of filtering, fSBI returns a pseudo-posterior $p_2(\theta | m_2)$, and setting $m.$ such that $\mathbb{I}_{g.}(m.) = \mathbb{I}(m.)$ to elide notation, we get:

$$p_2(\theta | m_2) = \frac{p(m_2 | \theta)\pi_1(\theta)}{p(m_2)} \tag{6.10}$$

$$= \frac{p(m_2 | \theta)p_1(\theta | \mathbb{I}(m_1)}{p(m_2)} \tag{6.11}$$

$$= \int_{\mathbb{I}(m_1)} dm_1 \frac{p(m_2 | \theta)p_1(\theta | m_1)}{p(m_2)} \tag{6.12}$$

$$= \int_{\mathbb{I}(m_1)} dm_1 \frac{p(m_2 | \theta)p(m_1 | \theta)\pi_0(\theta)}{p(m_2)p(m_1)} \tag{6.13}$$

$$\Rightarrow p_2(\theta | \mathbb{I}(m_2)) = \iint_{\mathbb{I}(m_1),\mathbb{I}(m_2)} dm_2 dm_1 \frac{p(m_2 | \theta)p(m_1 | \theta)\pi_0(\theta)}{p(m_2)p(m_1)} \tag{6.14}$$

Note that the *true* Bayesian posterior after two rounds would be, using Bayes' rule:

$$p_{\text{true}}(\theta | \mathbb{I}(m_2), \mathbb{I}(m_1)) = \frac{p(\mathbb{I}(m_2), \mathbb{I}(m_1) | \theta)\pi_0(\theta)}{p(\mathbb{I}(m_2), \mathbb{I}(m_1))} \tag{6.15}$$

$$= \int_{\mathbb{I}(m_1),\mathbb{I}(m_2)} dm_2 dm_1 \frac{p(m_2, m_1 | \theta)\pi_0(\theta)}{p(m_2, m_1)} \tag{6.16}$$

Eqns 6.14 and 6.16 would be equal iff the metrics $m_1$ and $m_2$ were conditionally independent. In other words:

$$p_2(\theta|\mathbb{I}(m_2)) = p_{\text{true}}(\theta|\mathbb{I}(m_2), \mathbb{I}(m_1)) \quad \text{iff} \tag{6.17}$$

$$p(\mathbb{I}(m_2), \mathbb{I}(m_1)|\theta) = p(\mathbb{I}(m_2)|\theta)p(\mathbb{I}(m_1)|\theta) \tag{6.18}$$

Since we do not explicitly take the correlations between metrics into account in fSBI, this indicates that this fSBI pseudo posterior is broader than the true posterior. In other words:

$$\int p_2(\theta|\mathbb{I}(m_2))d\theta > \int p_{\text{true}}(\theta|\mathbb{I}(m_2), \mathbb{I}(m_1))d\theta \tag{6.19}$$

Intuitively, sampling from the l.h.s. distribution in the equation above is less restrictive than sampling from the r.h.s.: we could potentially sample $\theta$ leading to simulations $x$ that respect either condition $a_1 < m_1 < b_1$ or $a_2 < m_2 < b_2$, but not both.

**New metrics in addition to metrics from previous rounds**: If we condition on all preceding metrics at every filtering round, we have a more restrictive distribution than the true posterior. Here, fSBI explicitly targets:

$$p_2(\theta|\mathbb{I}(m_2), \mathbb{I}(m_1)) = \iint_{\mathbb{I}(m_1),\mathbb{I}(m_2)} dm_1 dm_2 \frac{p(m_2, m_1|\theta)p_1(\theta|m_1)}{p(m_2, m_1)} \tag{6.20}$$

$$= \iint_{\mathbb{I}(m_1),\mathbb{I}(m_2)} dm_1 dm_2 \frac{p(m_2, m_1|\theta)p(m_1|\theta)\pi_0(\theta)}{p(m_2, m_1)p(m_1)} \tag{6.21}$$

$$= \iint_{\mathbb{I}(m_1),\mathbb{I}(m_2)} dm_1 dm_2 \frac{p_{\text{true}}(\theta|m_2, m_1)p(m_1|\theta)}{p(m_1)} \tag{6.22}$$

$$< \iint_{\mathbb{I}(m_1),\mathbb{I}(m_2)} dm_1 dm_2 p_{\text{true}}(\theta|m_2, m_1) \tag{6.23}$$

This extra factor $\frac{p(m_1|\theta)}{p(m_1)}$ in the ratio inside the integral collapses the neural density estimator to a narrower marginal density, since we now marginalize the true posterior $p_{\text{true}}(\theta|m_1, m_2)$ over $p(\mathbb{I}(m_1)|\theta)$ to estimate this density[2].

Intuitively the fSBI posterior mass is restricted strictly to the regions satisfying $\mathbb{I}_{g_1}(m_1)$. This is clearly more conservative than sampling from the true posterior jointly conditioned on all possible values of $m_1$ and $m_2$, and then marginalising over $\mathbb{I}(m_1)$ and $\mathbb{I}(m_2)$.

Thus, the former fSBI procedure allows us to find a broader density from which to sample parameters—this allows us to explore the space of possible parameters more thoroughly with respect to each metric, while having the disadvantage that by ignoring correlations between metrics, we can be sampling from regions of posterior space where conditions on both metrics are not satisfied. The latter approach is more conservative, such that we are highly likely to find samples that satisfy the condition on $m_1$, but at the cost of leaving many regions of parameter space unexplored.

---

[2]Note that we can correct for this simply by using importance weights $\frac{p(m_1)}{p(m_1|\theta)}$ in the integral

## 6.6 fSBI on polynomial search space

Table 4:

| Plasticity parameters, polynomial search space | | |
| --- | --- | --- |
| Learning rate | $\eta$ | 0.01 |
| Tunable synaptic traces time constants | $\tau_{XY}^{pre}, \tau_{XY}^{post}$ | $\in [10, 100]$ms |
| Other tunable plasticity parameters | $\alpha_{XY}, \beta_{XY}, \gamma_{XY}, \kappa_{XY}$ | $\in [-2, 2]$ms |
| Maximum weight (EE, EI, IE, II) | $w_{max}$ | 20 |

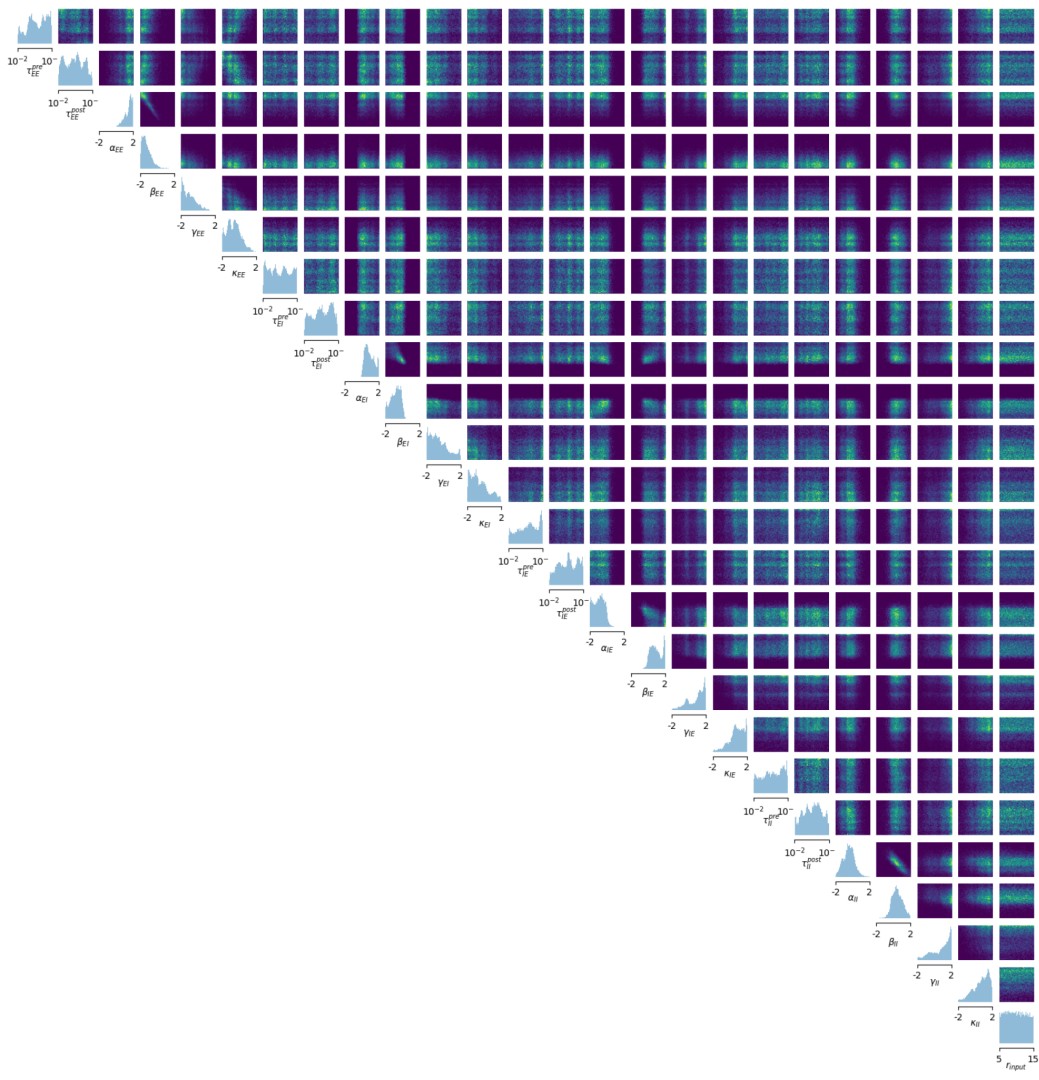

Figure 6.1: Pairplot of 10k plausible (obeying all conditions) rules from the polynomial search space. The diagonals show the distribution of values for each plasticity parameter. The off-diagonal plots show the marginals between each pair of plasticity parameters.

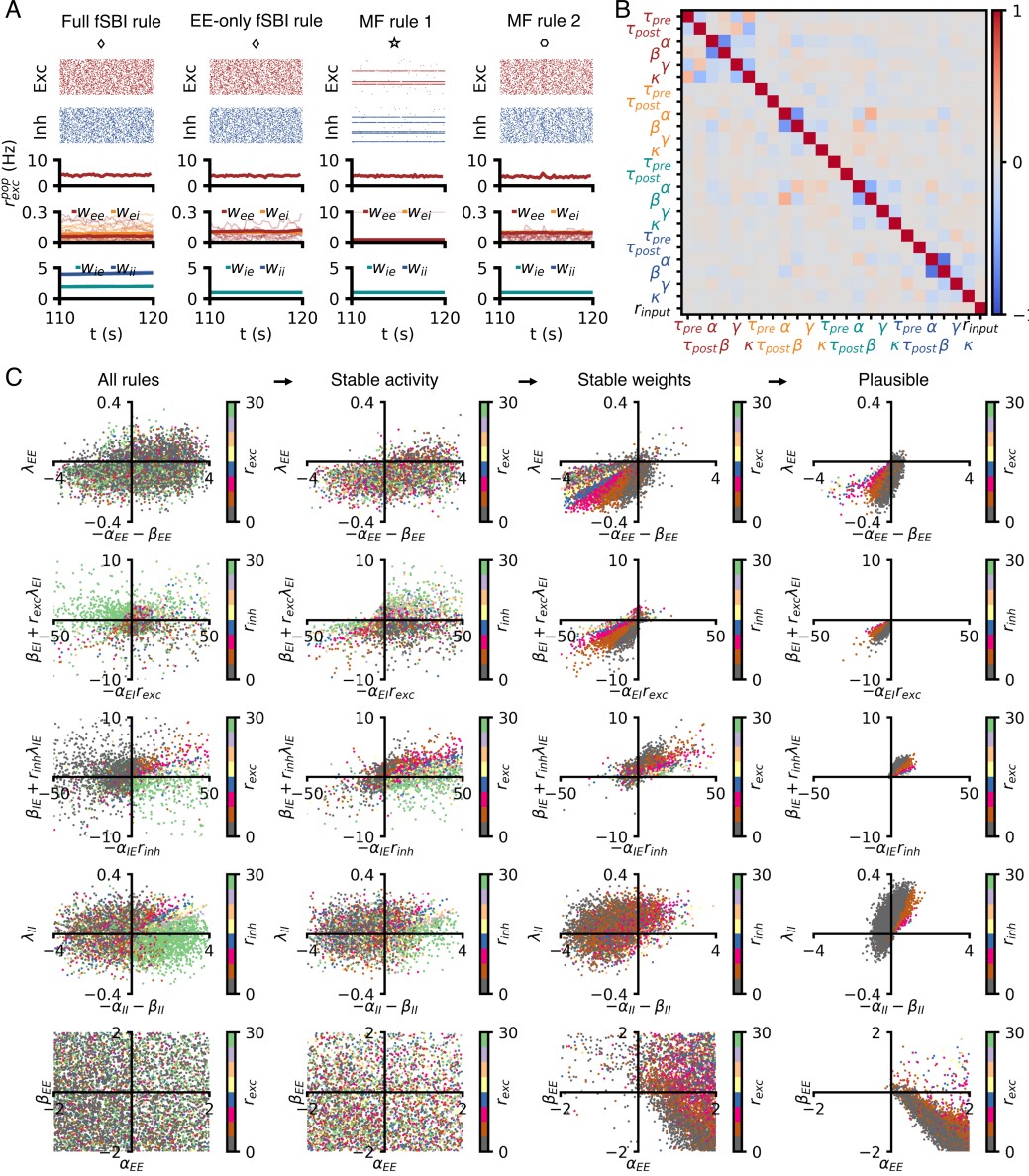

Figure 6.2: Companion figure to Fig. 3 **A**: Full visualization of the networks simulated with rules shown in Fig. 3D, E. **B**: Pearson correlation matrix of 10k plausible rules (obeying all conditions) from the polynomial search space. Less structure is seen than in Fig. 3A, since a wide collection of rules with various corresponding network dynamics (e.g., activities between 1 and 50Hz) is included. **C**: Emergence of mean-field-like structure for EE, EI, IE, and II rules, polynomial search-space.

## 6.7  fSBI on MLP search space

Table 5:

| Plasticity parameters, MLP search space | | |
|---|---|---|
| Tunable learning rate | $\eta_{EE}$, $\eta_{IE}$ | $\in [0, 1]$ |
| Tunable weights and bias | $\theta \in \{W_{pre}^{EE}, W_{post}^{EE}, W_{pre}^{IE}, W_{post}^{IE}\}$ | $\in [-1, 1]$ |
| Time-constant of $C_{exc}$ | $\tau_{Cexc}$ | 10ms |
| Time-constant of $C_{inh}$ | $\tau_{Cinh}$ | 100ms |
| Time-constant of $\langle V_j \rangle$ | $\tau_{Vtrace}$ | 100ms |
| Maximum weight (EE, IE) | $w_{max}$ | 20 |
| Fast pre-synaptic traces time-constant | $\tau_{pre\ EE}^{(1)}$, $\tau_{pre\ IE}^{(1)}$ | 10ms |
| Slow pre-synaptic traces time-constant | $\tau_{pre\ EE}^{(2)}$, $\tau_{pre\ IE}^{(2)}$ | 100ms |
| Fast post-synaptic traces time-constant | $\tau_{post\ EE}^{(1)}$, $\tau_{post\ IE}^{(1)}$ | 10ms |
| Slow post-synaptic traces time-constant | $\tau_{post\ EE}^{(2)}$, $\tau_{post\ IE}^{(2)}$ | 100ms |

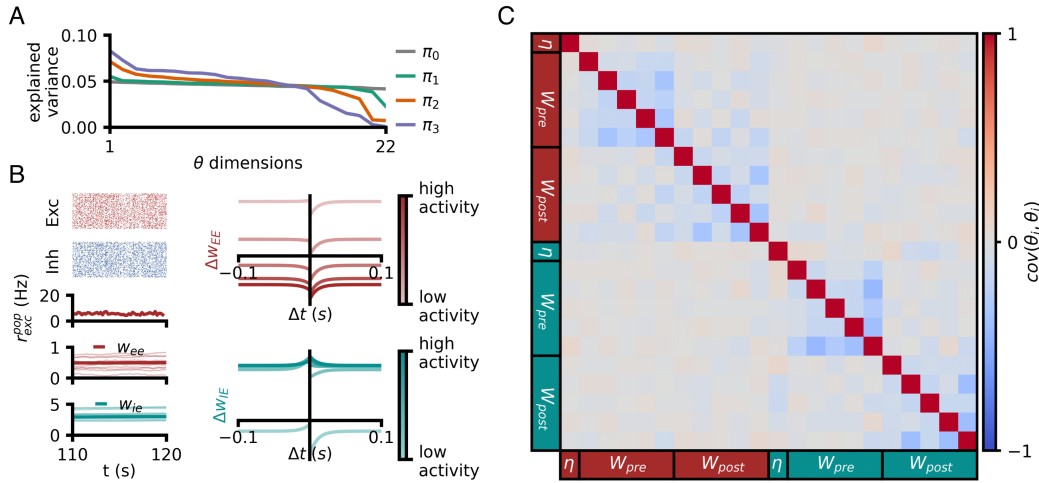

Figure 6.3: Companion figure to Fig. 4 **A**: Principal component analysis (PCA) on samples from each intermediate fSBI MLP posterior ($\pi_0 \to \pi_3$). Variance explained along the corresponding PCA directions across rounds. **B**: Another plausible rule from the MLP search space (same plot as in Fig. 4D) and its visualization under the pre-post protocol. **C**: Pearson correlation matrix between the rule parameters of 10k plausible rules from the MLP search space.

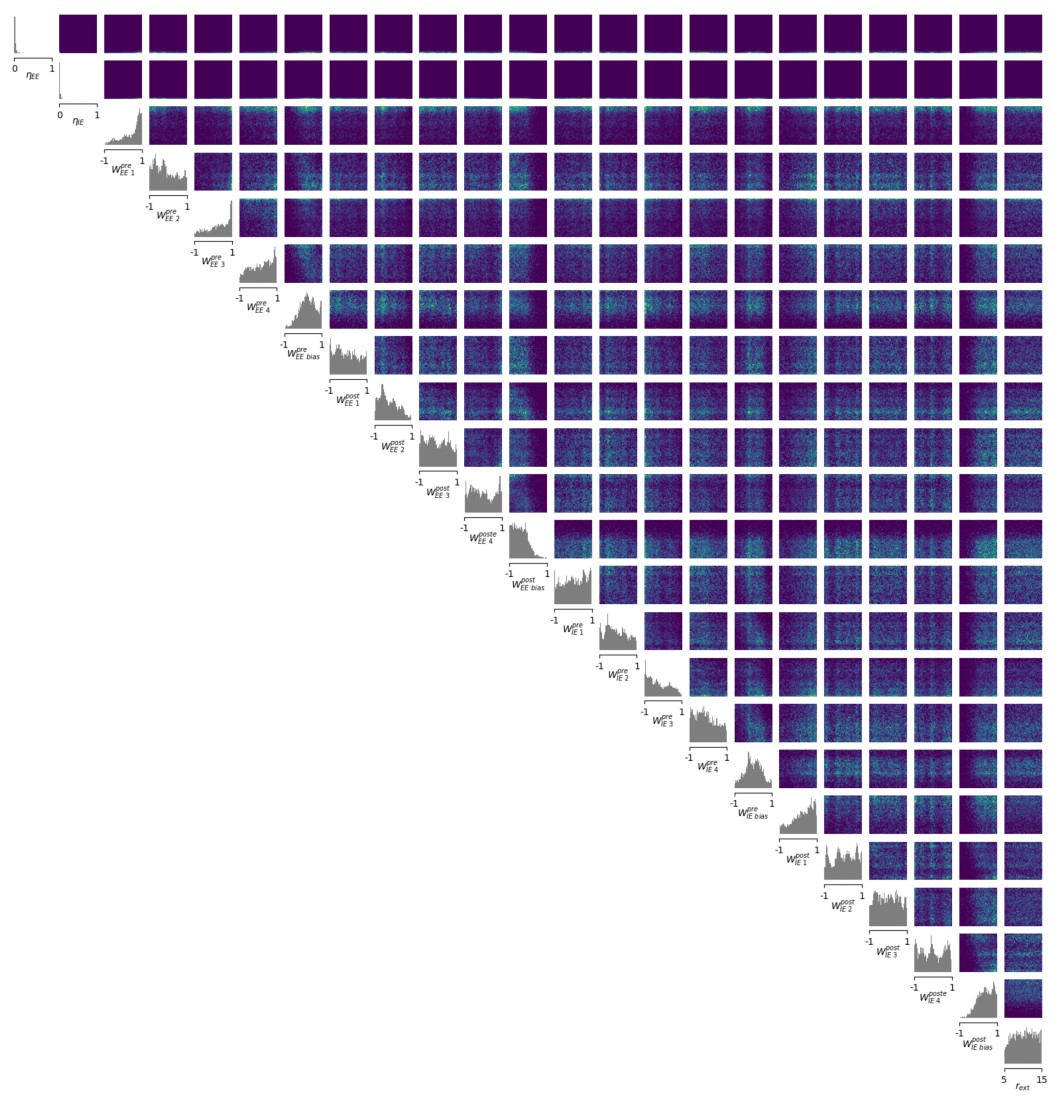

Figure 6.4: Pairplot of 10k plausible (obeying all conditions) rules from the polynomial search space. The diagonals show the distribution of values for each plasticity parameter. The off-diagonal plots show the marginals between each pair of plasticity parameters.

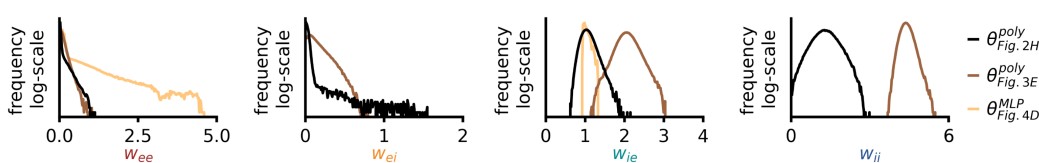

Figure 6.5: Weight distributions of some meta-learned rules.

**Proof of principle of the flexibility of the MLP search space**: Below, we detail a preliminary analysis suggesting that approximating potentially highly non-linear plasticity rules in spiking networks could be done using a partially trained MLP. Refer to Section. 2.2 for the definition and architecture of one MLP.

First, we trained the final layer of one MLP (corresponding to a "half rule", i.e., weight updates at a pre-synaptic spike, or at a post-synaptic spike, with the same shared weights as the ones used in Fig. 4) to approximate the weight updates on a pre-synaptic spike of the rule proposed by Vogels et al. [7]:

$$\frac{\mathrm{d}w_{ij}}{\mathrm{d}t} = \eta \delta_i(t)(-0.2 + x_j(t)), \tag{6.24}$$

Note that this rule belongs to our polynomial search space. This fit does not involve any network simulations, we instead minimize the loss $\mathcal{L} = \mathrm{MSE}(\Delta w_{\mathrm{MLP}}, \Delta w^*)$ with the standard auto-differentiation tools available in Pytorch. The result, shown Fig. 6.6, suggests that this linear rule can be well approximated by the MLP.

Finally, to check in practice how well-behaved this MLP search space would be, we used an evolutionary strategy (CMA-ES [58]) to find IE MLP rules that would establish a firing rate of 10Hz after 1min of simulation of a SNN similar to the ones described in Section. 2.1. The loss optimized was

$$\mathcal{L}(\theta) = \left\langle \frac{\left(\langle r_{\mathrm{exc}} \rangle_t - r_{\mathrm{target}}\right)^2}{\langle r_{\mathrm{exc}} \rangle_t + \epsilon} \right\rangle_{\mathrm{trials}}, \tag{6.25}$$

Each rule was evaluated on $n_{\mathrm{trial}} = 5$ trials. The evolution of the loss across meta-iterations (epochs of training the MLP plasticity rule with CMA-ES) shows that satisfying rules are found within a dozen iterations (Fig. 6.6B), suggesting that the search space is well-behaved enough to be navigated by ES optimizers such as CMA-ES. When simulating a learnt IE rule in a SNN, we see that the rule stabilizes the population firing rate at the desired target (Fig. 6.6C) after 2min of simulation. Note that the weight changes and values are not plausible, which is not surprising given this criterion was not part of the meta-objective. In contrast, fSBI was designed such that one can post-hoc introduce "regularizers" to an implicit loss, thus allowing to react on-the-fly to the implausible features of the considered candidate rules.

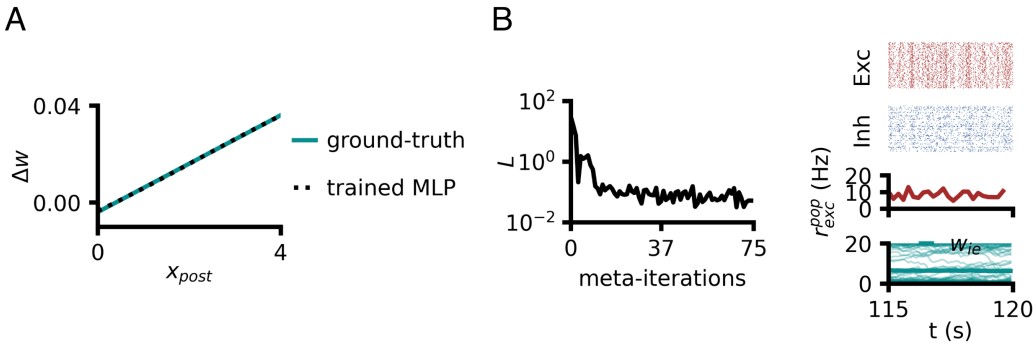

Figure 6.6: Proof of principle of the flexibility of the MLP search space. **A:** MLP with the same architecture and shared weights as the one in Fig. 4 trained to approximate the update at a pre-synaptic spike of the rule proposed in Vogels et al. [7]. **B:** SNN with MLP-parameterized rule for IE connections. The same parameters as in Fig. 4 of one MLP rule were trained with CMA-ES [58] in order to target a 10Hz population firing rate. Left: evolution of the loss function. Right: raster plots, population activity and IE weight traces of a SNN evolving with the meta-learnt solution (note that the loss did not include a penalty regarding the plausibility of weight traces).