# OpenReview forum: "Meta-learning families of plasticity rules in recurrent spiking networks using simulation-based inference"
_NeurIPS.cc/2023/Conference — NeurIPS 2023 poster_

### Official Review · Reviewer_bG1x · 2023-06-16

**Soundness:** 3 good
**Presentation:** 4 excellent
**Contribution:** 2 fair
**Rating:** 6
**Confidence:** 3

**Summary:**

The authors propose a novel meta-learning approach based on Bayesian filtering for discovering synaptic plasticity rules that entrain a E-I recurrent spiking neural network to satisfy a set of predefined criteria (e.g. the network firing rates occupy a reasonable regime at steady-state, with appropriate variability statistics, etc.). Using two different parameterizations of their synaptic plasticity rule (a polynomial plasticity rule resembling many different kinds of experimentally observed STDP vs. a multilayer perceptron parameterization) the authors show that they are able to recover many heterogenous rules that match their pre-specified target dynamical regimes, which in some cases align to predictions made by a mean-field model, and in other cases differ in interesting ways.

**Strengths:**

The paper is well written and very clear. Furthermore, the comparisons between the families of plasticity rules derived from mean-field analytic approximations and those derived from fSBI are very interesting--that the authors found plasticity rules that would not have been predicted to be effective under the mean-field approximation is an endorsement of the value of their method, and the fact that several similar-seeming plasticity rules were found to produce markedly different dynamical spiking regimes  has interesting implications for the interpretation of experimental results. The authors' emphasis on multiple different learning rules for different cell types is a welcome improvement over previous methods, and it is a further benefit of their approach that fSBI produces entire families of plasticity rules rather than requiring full commitment to any one particular learning rule (this may allow certain types of predictions across cell types which I elaborate on below).

**Weaknesses:**

Though the results are technically solid, thorough, and interesting enough to warrant publication in my opinion, I have serious doubts about the long-term value of this conceptual framework overall for several reasons that I think it would be very helpful for the authors to elaborate on.

First of all, the learning rules uncovered by fSIB do not actually do anything beyond stabilize the network in a reasonable dynamical regime. It is unclear whether these forms of plasticity can be used for any other interesting functions, such as memory formation or task performance. While this is not an unattainable goal for meta-learning approaches in general (see e.g. Confavreux & Vogels 2020), these alternative approaches use either gradient-based or in some cases evolutionary learning strategies which are likely better suited to searching in high dimensional spaces than sampling-based methods such as fSIB. Without learning to perform ecologically relevant tasks and without comparing to existing alternative methods, we are left wondering whether fSIB is actually useful as an optimization method in this space.

Second, and relatedly, the flexibility of the plasticity rules learned by fSIB (and most meta-learning alternatives) is limited by the imaginations of the algorithm designers & the scalability of the algorithm. For instance, there are a wide variety of plasticity algorithms that have been proposed for non-trivial task learning that would not be implementable within the parametric space proposed by the authors (e.g. reward-modulated plasticity, as the authors mention; burst-mediated learning as in Payeur et al. 2021; any form of learning involving dendritic compartmentalization as in Urbanczik & Senn 2014; or possibly even forms of phenomenological plasticity based on spiking triplets etc.). The search space is so wide, that all one may do is select a space that accommodates some known set of existing plasticity rules, which makes one wonder whether meta-learning methods for plasticity are in general doing little more than rediscovering algorithms that were already previously identified, but without the theoretical guarantees provided by algorithms motivated by optimization principles.

This second point would be less relevant if it were possible to generate genuine, testable predictions based on meta-learned plasticity algorithms that go above and beyond those already identified in existing literature. In this study, there appear to be some plasticity rules identified that are not already supported by the mean-field model. However, we are provided with a wide ensemble of potential plasticity parameterizations, and it's unclear what to actually do with this distribution. For instance, could it be used to infer, given a plasticity rule for one type of neuron (e.g. EE synapses), a probability distribution over the parameters of the other synapses (EI, IE, II)? Should we expect more variability in plasticity parameters in biology in dimensions of parameter space that are not constrained by the mean-field equations for stabilizing the network?


**Questions:**

The scaling of the algorithm doesn't seem to be analyzed much beyond one parameterization each for the polynomial and MLP cases which work effectively. Does the MLP method scale well as the number of optimized parameters increases?

How well do the learned plasticity rules generalize to any of the following: to different network architectures (e.g. multiple layers or different connectivity probabilities), to changes in neuron properties (e.g. excitability or spiking properties), or variations in input statistics?

**Limitations:**

There are no obvious potential negative societal impacts of the proposed work, and the authors do address many of the limitations of their work (I mention a few that they do not address above).

---

> ### Author Rebuttal · Authors · 2023-08-09
>
> We thank the reviewer for their insightful review. We have already addressed the more major concerns above, but elaborate here on the more technical and philosophical points that we couldn’t cover in the common rebuttal:
>
> ### 1/ Local optimization vs fSBI:
>
> The reviewer raised the point that sampling methods such as fSBI may be ill-suited to find precisely-tuned, functional rules. Our response to this concern is multi-pronged. First off, in the unlikely event (see below) that fSBI fails to find at least some finely tuned rules, the rules found here may still provide interesting initializations for local optimization.
>
> However, we believe that it is implausible for plasticity rules to be finely tuned, as such fine orchestration would likely lead to catastrophic network-level consequences in the face of perturbation. What’s more, based on current understanding, noisy biochemical processes and sensors could not implement them. We also believe that plasticity rules are unlikely to be unique, given that degeneracy is widespread in biological systems (Prinz et al., Nat. Neur. 2004). These arguments suggest that there should be many functionally interesting rules, instead of a unique solution that a sampling method such as fSBI would indeed struggle to find. Such speculations are supported by the (preliminary) evidence in Fig.R2 that many rules selected for plausibility ascribed a function to a network during a familiarity detection task.
>
> To provide some context, we turned to SBI because of the difficulty of scaling meta-learning approaches such as those presented by Confavreux et al. NeurIPS 2020 or Jordan et al. eLife 2021 —which consider a few spiking neurons and no recurrence— to large recurrent spiking networks with more flexible plasticity search spaces. In our hands, the added expressive power of such networks and rules made defining a satisfying loss function very difficult. Specifically, we could meta-learn plasticity rules that solved various specific tasks, but the networks would lack many biologically plausible features, instead displaying synchronous and/or regular activity, unrealistically fast weight changes, or unrealistic distributions of firing rates. We could address some of these problems by adding terms to our loss functions, but every metric had blind spots (see main paper Fig.2E,F,G) which meant adding (and having to tune) more and more terms in the loss, which quickly became untenable. Hence we decided to first filter out all the rules that were not producing plausible networks, flexibly, one constraint at a time, implicitly crafting a comprehensive loss function on the fly, without the need for fine-tuning.
>
> We would also like to emphasize that Confavreux et al. don’t investigate any function beyond stability in their spiking neuron experiments.
>
> ### 2/ Meta-learning vs analytical approaches:
>
> The reviewer raised the concern that meta-learning might not have the potential to discover truly novel rules. We believe that our results answer this question as we found rules that are not predicted by mean field assumptions. Of course, we believe that both approaches must complement one another, and our results may inspire refinements to mean-field theory to allow for the inclusion of discovered rules. Conversely, the choice of the plasticity search space should also be informed by analytical studies proposing promising plasticity mechanisms.
>
> ### 3/ Extending the plasticity search space, scaling of fSBI:
>
> The reviewer suggested extending our study to multi-compartmental models and correspondingly more detailed plasticity models. This is an interesting avenue for future work, but unfortunately out of scope. However, in line with this idea, we highlight that bursts have been implemented previously as high powers of synaptic traces (Zenke et al. Nat. Comm. 2015). Thus, our MLP search space could, in theory, detect and use bursts to craft weight updates.
>
> Considering the scaling of the method with the number of parameters of the plasticity search space, it is computationally expensive to change search space, hence our focus on two search spaces. Previous applications of SBI were successful up to 30 parameters (Cranmer et al. PNAS 2020, Goncalves et al. eLife 2020), and recent developments have increased this bound in specific settings to a few 1000 parameters (Ramesh et al. ICLR 2022). One of our proposed fixes for this dilemma was to keep some of the MLP parameters fixed, so the effective dimension is low, while the rule search space dimension remains as high as possible. The definition of a flexible yet moderately high-dimensional search space is a key step that will need much more work. This limitation is shared with other meta-learning approaches (Confavreux et al. NeurIPS 2020, Jordan et al. eLife 2021), but we believe we make a valuable contribution towards this goal.
>
> ### 4/ What to do with the distributions of rules proposed:
>
> We agree with the reviewer that unraveling and interpreting the structure we found in the posteriors is a major extension of our work, that is out of scope of this study. However, we are currently working on methods to make sense of the high-dimensional posteriors that come out of fSBI, building on previous studies (Deistler et al., PNAS 2022; Gonçalves et al. eLife 2020). The reviewer’s suggestion to fix one rule (e.g., EE) and look at how it constrains the other rules is a possible analysis (conditional posteriors) and an interesting avenue for future work.
>
> ### 5/”Should we expect more variability in plasticity parameters in biology in dimensions of parameter space that are not constrained by the mean-field equations for stabilizing the network?”
>
> This is an interesting suggestion, and we have begun investigating this, but we currently don’t have clear answers. We will discuss this in the paper, space permitting.

---

> > ### Comment · Reviewer_bG1x · 2023-08-10
> > **Reply to authors**
> >
> > Thank you for your response to my review. Your responses are adequate, and I believe the paper will be improved by incorporating the discussion. I will maintain my score, which was already above the threshold for acceptance.

---

### Official Review · Reviewer_jNn6 · 2023-06-30

**Soundness:** 3 good
**Presentation:** 4 excellent
**Contribution:** 3 good
**Rating:** 6
**Confidence:** 3

**Summary:**

The authors present the filter Simulation-Based Inference (fSBI) method, aimed at automating the discovery of families of rules that establish plausible dynamics within large spiking networks. This study seeks to overcome the challenges presented by human intuition and the overly simplified assumptions that have characterized prior research in synaptic plasticity—a critical element in brain-based learning and memory processes. The fSBI method facilitates the extensive exploration of the potential rule space, identifying sets of rules via metrics that effectively constrain network dynamics to satisfy the intended conditions. Importantly, this approach evaluates one metric at a time, significantly reducing computational complexity compared to simultaneous evaluation of multiple metrics.

Overall, this study provides a principled approach for meta-learning a family of plasticity rules in networks, offering a starting point for deeper analysis of computation and plasticity mechanisms. Additionally, the study highlights that classical experimental predictions often yield contradictory results when considered in isolation, thereby challenging traditional theoretical and experimental protocols for understanding plasticity. Furthermore, the inherent flexibility of this tool lends it the potential to incorporate new neuroscience findings into the metrics and architecture as emerging data flows in.


**Strengths:**

o	The manuscript is very well-written, with informative illustrations.

o	The proposed fSBI framework exhibits flexibility across architectures and rule parameterization, and can be applied in unsupervised learning contexts.

o	The viability of the learned rules and their resulting network-level effects is theoretically grounded based on mean-field predictions.

o	Analysis pertaining to the reduction in effective dimensionality of rule parameters and interdependence of plasticity rule mechanisms is insightful.


**Weaknesses:**

o	Despite only evaluating one metric at a time, the method may still prove computationally demanding if the number of rule parameters is significant, to accommodate a broader rule basis. The authors recognize this "curse of dimensionality", and the Appendix notes simulations took two months.

o	A minor point, but the paper could benefit from more detailed explanation on how specific values for the metric bounds were chosen and how sensitive these choices affect the results (in addition to what's already in 5.2).

o	I could not find information on whether simulations were repeated across different initializations and seeds.


**Questions:**

o	Although the study acknowledges that slower homeostatic mechanisms aren't considered, how might the inclusion of weight regularization (or homeostatic mechanisms) influence the pool of plausible rules and the interaction between plasticity terms?

o	Neuromodulation is not accounted for in this study, but could the authors briefly discuss (perhaps in Discussion) the feasibility of extending this method to infer rule parameters in recently proposed modulated-eligibility trace-based biologically plausible learning rules for training recurrent spiking neural networks [1-3]?

o	Could the authors provide their intuition/hypotheses on potential reasons why rules appearing similar/different under pre-post protocols can result in different/similar network-level effects?

o	What does the resulting weight distribution look like for the plausible candidates? Is it lognormal?

o	References:

	[1] Guillaume Bellec, Franz Scherr, Anand Subramoney, Elias Hajek, Darjan Salaj, Robert Legenstein, and Wolfgang Maass. A solution to the learning dilemma for recurrent networks of spiking neurons. Nature Communications, 11(1):1–15, dec 2020.

	[2] James M Murray. Local online learning in recurrent networks with random feedback. ELife,8:e43299, 2019.

	[3] Yuhan Helena Liu, Stephen Smith, Stefan Mihalas, Eric Shea-Brown, and Uygar Sümbül. Cell-type–specific neuromodulation guides synaptic credit assignment in a spiking neural network. Proceedings of the National Academy of Sciences, 118(51), 2021.


**Limitations:**

The authors have acknowledged various limitations: the curse of dimensionality associated with fSBI, limited interpretability of MLP, the use of noise-free simulations, not considering long-term stability measures, among others. These limitations might impact the study's conclusions, including the pool of plausible rules and the nature of interactions between plasticity terms. However, despite these constraints, the study introduces an encouraging framework (fSBI) for inferring plasticity rules that could be enhanced in future research to incorporate additional considerations.

---

> ### Author Rebuttal · Authors · 2023-08-09
>
> We thank the reviewer for their positive assessment of our method and its usefulness.
>
> ### 1/ Compute requirements:
> While we agree that this method is not computationally cheap, this is also true of other meta-learning methods. Notably, our study can be considered as a form of pre-training of plasticity rules, which only needs to be performed once, and then used as a starting point for various investigations into more specific functions.
>
> ### 2/ Initialization/seeds:
> Every rule sampled was simulated once in a network with a new initialization/seed (affecting input strength, connectivity, and the timings of Poisson inputs, denoted as nuisance parameters). Due to the sequential nature of fSBI, robustness to these factors was enforced implicitly, across rounds. The plastic network dynamics shown in Fig 2.H, 3D,E 4D of the main paper were simulated on unseen initialization/seed/input rate, confirming the robustness of the method.
>
> ### 3/ Weight regularization/homeostatic mechanism:
> The MLP rules have a dependence on the current value of the synaptic weights. Some studies propose plasticity rules that are stabilized by considering all the outgoing or incoming weights of a neuron (e.g. Litwin-Kumar 2014, Schulz et al. 2021). It would be an interesting extension for future work to add such terms to the plasticity rules.
>
> ### 5/ Neuromodulation and reward-based learning rules:
> This is an interesting extension to the study, and possible in principle (but out of scope on the short time scale of this review). It either involves defining a new search space and applying fSBI from scratch, or if we consider that reward modulates the learning rule, it may be possible to add a term to the existing search spaces and start from the distributions of rules proposed in this study.
>
> ### 6/ Why rules with similar pre-post appear different at the network level:
> The pre-post protocol only considers a pair of a pre- and a postsynaptic spike, i.e., two spike-triggered weight-updates. For the polynomial search space, this means that the weight change of a protocol is $\Delta w = \alpha + \beta + \kappa e^{-\Delta t/\tau_{post}}$ for (post-then-pre pairs), and $\Delta w=\alpha + \beta + \gamma e^{-\Delta t/\tau_{pre}}$ for pre-then-post pairs. For instance, this protocol does not constrain the exact values of the non-Hebbian terms, $\alpha$ and $\beta$, only their sum. So two rules with different mechanisms — e.g., one with no on-pre depression ($\alpha=0$), and the other with a non-zero $\alpha$, compensated by on-post depression — may appear identical under the pre-post protocol. However, their underlying parameters, mechanisms, and corresponding network behavior may be very different. In the MLP, there are even more synaptic variables and parameters, which are also unconstrained or partially constrained by the pre-post protocol. This adds even more degeneracy in the rules that could appear identical/similar under the pre-post protocol.
> Note that the analytical considerations above seem to echo experiments: experimental data on these protocols are notoriously difficult to reproduce because the result depends on other experimental variables, e.g., frequency of the spike-pairings, other inputs to the neurons, or history of the connection.
>
> ### 7/ Resulting weight distributions:
> We did not add metrics requiring weight distributions to be lognormal [10], though this would be an interesting future step towards biological plausibility, and we may be able to add it before the paper deadline. The shape of the resulting weight distributions depends on the rule and the synapse type. We have added the weight distribution of the three example rules shown in the main text in the revised Fig.1E. Because we prevent too many weights from blowing up (metrics on stable weight trajectories), we do not expect unnatural, bimodal distributions like in the case of traditional Hebbian learning. We note that when receiving more specific inputs (solving tasks), these distributions may change, so the rule is not the only determinant of the final weight distribution.

---

> > ### Comment · Reviewer_jNn6 · 2023-08-13
> > **Response acknowledged**
> >
> > Thank you for addressing the points I raised in my review. I'm confident that the paper will benefit from the included discussions. My initial rating, which was favorable for acceptance, remains unchanged.

---

### Official Review · Reviewer_5no9 · 2023-06-30

**Soundness:** 3 good
**Presentation:** 3 good
**Contribution:** 3 good
**Rating:** 7
**Confidence:** 3

**Summary:**

The manuscript introduces a novel way of sampling plasticity rules that obey certain plausibility constraints. Specifically, the authors focus on networks of excitatory and inhibitory spiking neurons and use a simulation-based inference (SBI) technique to filter plasticity rules based on the behavior of the networks in which they are implemented. They demonstrate the effectiveness of the technique, and in particular find that standard neuroscience protocols for examining synaptic plasticity are insufficient for identifying plasticity rules.

**Strengths:**

*Originality:*
This is a nice extension of simulation-based inference that significantly reduces the size of the search space by enforcing constraints one by one instead of all at the same time.

*Quality:*
The work seems carefully done, with a significant number of simulations to support their conclusions.

*Clarity:*
The presentation is clear and the paper is well-written.

*Significance:*
Understanding the plasticity rules used by neural circuits is of utmost importance in neuroscience. This paper addresses this question in an interesting way, and the authors show shortcomings with standard neuroscientific approaches.

**Weaknesses:**

There are a number of choices related to the simulations that seem insufficiently discussed: e.g., the ratio of excitatory to inhibitory neurons in the networks that were simulated; the sizes of these networks; the statistics of the inputs to the networks; the initial weight distribution; the duration of the simulations; the various numerical constants used in the metrics from section 2.4.

Also, while the authors point out that standard protocols from neuroscience do not uniquely pin down bio-plausible plasticity rules, it would be good to at least speculate on possible approaches that could solve this problem.

**Questions:**

1. Do we need to worry about any biases in the sampling obtained using fSBI?  If there are such biases, would they depend markedly on details such as the order in which the constraints are imposed?
2. How were the details of the simulations mentioned above (like size of networks) chosen? Were multiple choices explored and the choice did not matter? Were some of the choices informed by measurements in real brains? If so, which areas and which animals -- and can we expect the results to generalize?
3. On line 82 it is written that *"For the MLP plasticity rule, only EE and IE synapses undergo spike-triggered updates"*. Why? Is this an arbitrary choice, or is there something deeper about it?
4. The comment on lines 268–270 suggests that many of the MLP rules are not, in fact, realistic. Should we then not try to either simplify the neural nets or add more constraints to restrict ourselves to realistic rules?

Minor comments:
* on line 86, what are the superscripts (1), (2)? are they the same as *pre*, *post*?
* on line 195, *"the order of the least to the most restrictive metric"*: how did you know which metrics were more restrictive?
* Figure 3, panel A: what do the colors for the axis labels represent?
* Figure 3, panels D, E: which side of the color bar is for small distances and which is for big?
* on lines 239-240: *"fSBI rules lead to plausible network dynamics both when simulated with only EE parameters [...] and also when using the rule for all connection types"*: why would we expect / want the dynamics to be stable when the plasticity rule applies only to some of the connections in the network?
* line 257: should mention the activation function(s) used for the MLP

**Limitations:**

The authors have adequately discussed limitations of their work.

---

> ### Author Rebuttal · Authors · 2023-08-09
>
> We thank the reviewer for the thoughtful review, please find below our attempt to address the points raised that are not in the common rebuttal.
>
> ### 1/ Biases in fSBI:
> The reviewer's concern about biases of the method and the influence of the filtering order is valid, as fSBI may suffer from the same biases as other simulation-based inference (SBI) algorithms: poorer posterior estimation due to limited training samples, curse of dimensionality, etc, which we discuss in the main paper.
>
> Biases may also be introduced by the fact that fSBI does not target the true Bayesian posterior: for a detailed discussion, we refer the reviewer to supplementary section 6.5. In particular, the constraints are likely not conditionally independent given the parameters, and thus the ordering may affect the posterior estimate. In our case, a less restrictive constraint on the rule space came before a more restrictive one, for sample efficiency. However, the restrictiveness of a rule and the corresponding fSBI posteriors are determined by a multitude of factors, such as the parametrization of the plasticity rule, network size, etc. We expect that biases introduced by constraint ordering, and indeed the ordering itself must vary on a case-by-case basis. Defining a rigorous method for ordering the constraints would require work beyond the scope of this paper.
>
> ### 2/ Simulation parameter choices:
> Besides the points made in the common rebuttal, we would like to add that the simulation parameters (network sizes, neuron model parameters, connectivity, initialization, inputs) were chosen to be as consensual as possible in the spiking network community. The parameters broadly emulate mammalian cortical networks and are taken from classical studies on dynamical systems analyses of static recurrent spiking networks (Brunel 2000, Dayan & Abbott 2001 for review, Vogels et al. 2011 and Sanzeni et al. 2023). We have tested the generalisability and robustness of our results on some variations of all of these parameters, with encouraging results.
>
> ### 3/ Spike-triggered MLPs:
> This was mostly a compute choice: running an entire MLP forward at every synapse and every integration timestep is prohibitively compute-intensive. It would be an interesting avenue for future research if a more efficient implementation than ours (that favored simulating many spiking networks in parallel) is possible.
>
> ### 4/ “Many of the MLP rules are not realistic”:
> The MLP rules are more difficult to interpret and we agree that understanding the structure of the plausible rules to make experimental predictions beyond the pre-post protocols is an important avenue for future work. For some specific values of synaptic variables that are not constrained by the protocol, the MLP rules may look like experimentally observed rules. Overall, given how underconstrained these rules are under the pre-post protocol, we do not think that this protocol is a meaningful test for these rules. However, as these rules combine biologically plausible synaptic variables into weight updates that amount to plausible network dynamics, we believe they are a meaningful description of a plasticity rule and have the potential to inform future experiments. We will clarify this in the paper.
>
> ### 5/ Speculate on which protocols may better disambiguate plasticity:
> This is an interesting avenue for follow-up work on our study, and we are currently exploring with some experimental collaborators. However, we have no tangible results to present yet. We believe that indirect data of plasticity rules such as activity dynamics and network behavior, which are more accessible experimentally, will help narrow down the potential plasticity rules at play.
>
> ### Minor comments:
> “on line 86, what are the superscripts (1), (2)? are they the same as pre, post?”: Each neuron has two eligibility traces with different timescales, inspired by the fast and slow eligibility traces in the triplet rule (Pfister et al. 2006). So x^(1)_i refers to the eligibility trace with time-constant 1 for neuron i.
>
> “how did you know which metrics were more restrictive?” We computed the fraction of rules that obeyed each metric condition (see Fig 2.C, 4.C) under the posterior estimate at a given inference round. This was taken as a proxy for the condition that was the easiest to fulfill in the next round of inference.
>
> “which side of the color bar is for small distances and which is for big?” This is an oversight from us, apologies. We will put back the scale in the published version: the darker, the higher the distance.
>
> “why would we expect/want the dynamics to be stable when the plasticity rule applies only to some of the connections in the network?” Each synapse type has its own rule in the polynomial search space, despite many previous theoretical studies only considering plasticity at EE synapses, and sometimes at IE synapses as well. We wanted to show that our sets of rules, even though technically sets of 4 co-active rules in a single network, are still plausible when taken in isolation.
>
> “the activation function(s) used for the MLP” is a sigmoid non-linearity. We added that information in the manuscript.

---

> > ### Comment · Reviewer_5no9 · 2023-08-11
> >
> > Thanks for the response! I believe the added discussion about the choice of metrics as well as the issues of robustness significantly improve the paper, so I will increase my score.

---

### Official Review · Reviewer_vJPD · 2023-07-28

**Soundness:** 3 good
**Presentation:** 3 good
**Contribution:** 3 good
**Rating:** 7
**Confidence:** 4

**Summary:**

How neurons update their synaptic strengths is an important and active area of research in neuroscience, with potential applications in ML. Much work has been done using analytically tractable mean-field models, in large part because the search space for possible rules is intractably large.  Here, the authors introduce an approach that cuts the search space down significantly.  While simulation-based inference has been used to infer parameter distributions that lead to network dynamics, the authors add constraints on plausible network dynamics to further reduce the search space to a manageable size.

They find that plasticity rules predicted by the mean field model they derive do not always result in plausible dynamics in the full network.  Moreover, pre-post diagrams that are often used in STDP analyses were found to be of limited use: rules with very similar diagrams could drastically diverge in terms of network dynamics, while rules with very similar network dynamics could result in very different pre-post diagrams.

**Strengths:**

Overall, this is a good study that contributes to the field. Brute force strategies are inherently problematic; the authors' filtering approach provides a helpful middle ground between approaches based on more tractable methods and full brute-force grid searches.  The comparison of the rules between the full and mean-field networks, as well as the findings of the pre-post diagrams, add nuance to the study of plasticity in computational and experimental settings.

**Weaknesses:**

Major:

The filter constraints are hard boundaries that are not always obviously well-founded.  For example, it isn't clear why firing rates above 50 Hz are considered implausible when population gamma oscillations typically go up to ~80 Hz and neurons have been found to fire up to ~1000 Hz. Why the bounds for $<w_{EX}>$ and $<w_{IX}>$ are an order of magnitude apart is unclear, nor is there physiological comparison (weights are in arbitrary units). Other bound decisions are similarly left ambiguous.

If the authors believe there are good reasons to have chosen the specific values presented, they should explain and cite the relevant work.  Otherwise, the paper should be rewritten to soften the claims.  In particular, the authors could instead choose to indicate that others might choose somewhat different bounds to be more or less inclusive, but that this is a first attempt and that the main conclusions are left unaffected.  The "plausible" network dynamics claims should thus be de-emphasized in that case.

Minor:

The MLP plasticity rule was introduced in a manner that was difficult to understand.  However, I found my questions answered when the results were introduced.  In particular, lines 252-264 should either be moved to line 82, when the rule is introduced (or integrated into that section), or vice versa.

The color bars in Fig 3D-E should have scales; if the lower bound is not very near zero, some of the associated claims may need to be softened.

Line 84: $C_i^I$ should be $C_i^{\mbox{inh}}$.

**Limitations:**

Yes

---

> ### Author Rebuttal · Authors · 2023-08-09
>
> We thank the reviewer for the constructive feedback, and for noting the potential of the approach. Please find a detailed response to the points that are not addressed in the common rebuttal below.
>
> ### Choice of metrics and their plausible ranges:
>
> We addressed most of this point in the common rebuttal above. Concerning the definition and interpretation of the weights in our model, weights affect the membrane potential via the driving force (membrane potential minus reversal potential, conductance-based model), which is different between excitatory and inhibitory synapses. This, in addition to the difference in the numbers of excitatory and inhibitory neurons, leads to requirements of larger inhibitory weights to achieve a balanced state, or close to ( Sanzeni 2023, Brunel 2000). In addition, synaptic weights are unitless because they are in units of leak conductance (10nS), but still comparable to experimental data via their potentials (PSPs), as mentioned in the common rebuttal.
>
> The lower bound of 1Hz was chosen mainly for numerical reasons: allowing us to collect enough spikes in a given 2min simulation and thus meaningfully assess rules with spike-triggered updates.
>
> In addition, we chose the ranges for the less widespread metrics, such as $\langle \sigma \rangle_i, \langle S \rangle_{i, f}, \langle Fano \rangle_i, \langle \rho \rangle_{i, t}, \langle Fano \rangle_t, w_{creep}, f_{w_{blow}}$, by benchmarking the respective metrics on example networks such as Poisson spike trains at various rates, networks from classical studies in the field (Vogels et al. 2011, Zenke et al. 2015, Litwin-Kumar & Doiron 2014) and also implausible networks —e.g., synchronous-regular regime.
>
> $f_{w_{blow}}$ was introduced to reject biologically unrealistic rules that led more than 10% of the weights of any synapse type to 0 or to the maximum weight $w_{max}$.
>
> $w_{creep}$ was designed to reject weights that changed unrealistically fast, excluding mean relative changes of more than 5% over the final 10s of simulation.
>
> We will make these motivations clear also in the manuscript.
>
> ### Minor comments:
> We will rephrase how the MLP rules are introduced in line 82 according to feedback.
>
> Colorbar Fig 3D,E. This was an oversight, apologies. Black corresponds to large distance and yellow to 0. We will add the legend in the final version.
>
> We will correct the typo line 84.

---

> > ### Comment · Reviewer_vJPD · 2023-08-21
> >
> > Thank you for the clarifications, especially regarding the chosen bounds, along with the changes the authors have indicated they will make to the manuscript.  I have increased my score.

---

### Author Rebuttal · Authors · 2023-08-09

We thank the reviewers for their constructive comments and their positive appraisal of our study, calling it "a good study that contributes to the field" (r. vJPD), "a nice extension of simulation-based inference" (r. 5no9), with "insightful” analysis” (r. jNn6), and the discovery of non-mean-field predicted rules deemed "an endorsement of the value of their method” (r. bG1x), and for unanimously recommending acceptance. Below, we discuss the most common questions and comments that have led to additional results and analyses (1 additional main figure and 1 supplementary figure below). We provide further details and additional responses in each individual review.

## 1/ Choice of metrics and their ranges:

Several of the reviewers commented on subpar reasoning for choices of parameters. Our choices were  informed by –and in line with– a range of modeling and experimental cortical studies [1-13], but we have now realized this was not sufficiently explained. We will add a more detailed discussion to Appendix 5.2. Key points:

Population firing rates $r_{exc}, r_{inh}$ were constrained between 1Hz and 50Hz, in line with a large swatch of literature reporting cortical firing rates of a few Hertz on average [1,8,9]. Notably, we only constrained population averages, individual firing rates could be higher or lower (r. vJPD).

Maximum average weights $<w_{EE}>, <w_{EI}>, <w_{IE}>, <w_{II}>$ were chosen so that PSPs were smaller than ~3 millivolts, a reasonable max weight according to experimental data [8, 10] (r. vJPD). Similarly, the other weight metrics ($f_{w_{blow}}, w_{creep}$) were chosen to reject biologically unrealistic scenarios, e.g., too many 0 weights or maximum weight $w_{max}$, or weights that changed unrealistically fast.

Cortical activity is broadly asynchronous-irregular [1,2,8,9,4]. Ranges for metrics assessing synchrony/regularity were thus devised with independent Poisson spike trains in mind, with leeway for spatiotemporal correlations, as seen in cortical recordings [11,12,13].

We will also make it more apparent in the manuscript that our choices of metrics and ranges are not universal, and a focus on specific brain regions, especially subcortical ones, may necessitate some changes  (r. vJPD). We’d like to highlight that fSBI allows for such changes of metrics and corresponding ranges post hoc, at little computational cost, because we store all spike trains and weight traces (and will make them accessible to the community).

## 2/ Robustness of fSBI rules:

In response to questions regarding the robustness of our results, we ran additional simulations and found that the “plausible” rules from the main paper generalized to different network sizes, sparsity, weight initializations, and ratios of E to I (Fig.R1A, r. 5no9 \& bG1x), although the respective metrics $f_{w_{blow}}$ and $w_{creep}$ were sometimes outside the predefined ranges.

We then tested “plausible” rules on 20min long simulations (versus 2min in the main paper, r. 5no9). Approx. 25% of the polynomial rules still met the conditions on all 15 metrics after 20min (Fig.R1B). Most other rules were  disqualified by our $<w_{IE}>$ and $<w_{II}>$ conditions, which select long-term stable weights, and are thus sensitive to simulation duration. Correspondingly, the II and IE plasticity parameters appeared more refined compared to the starting set (Fig.R1D). Overall, this suggests that the 2min task had properly constrained the EE and EI rules, but was too short to fully constrain the II and IE rules.

Similarly, 35% of the 2min-plausible MLP rules met all conditions in the 20min task. Interestingly, dimensionality reduction of the successful plasticity parameters did not reveal sub-structures, suggesting that the 20min task did not dramatically refine the set of filtered rules (Fig.R1C).

We thank the reviewers for their suggested controls and will add them to the paper. We believe that these experiments further emphasize the advantages of fSBI and its robustness.

## 3/ Rules with a specific network function:

Reviewer bG1x pointed out that it was unclear whether the candidate rules obtained in this study had the potential to ascribe any interesting network function beyond stabilizing a recurrent spiking network. We thus tested our “plausible” rules on a familiarity detection task (Fig.R2A).

We first tested which of the pre-identified, stable rules met the plausibility conditions during the familiarity detection task (over 80%, further confirming the rules’ robustness, Fig.R2C). Next, we introduced new metrics that were specific to the task, such as measuring the population activity during novel/familiar stimulus presentation. Novel and familiar stimuli were of identical magnitude, thus eliciting the same population response in the absence of plasticity. Interestingly, most polynomial rules responded more strongly to novel stimuli after learning (Fig.R2B), while MLP rules exhibited a more balanced distribution of preferences towards familiar, novel, or neither (Fig.R2D). Overall, many polynomial and MLP rules filtered for plausibility conferred a function —novelty or familiarity detection— to the network they were embedded in. Our new results indicate that rules with interesting network functions beyond mere stabilization may not be so rare in the set of filtered rules presented in this study.

## References:
[1] Dayan and Abbott, 2001

[2] Brunel, Jour. of Comp. Neur., 2000

[3] Vogels, Rajan and Abbott, Ann. Rev. Neur., 2005

[4] Renart, …, Harris, Science, 2010

[5] Vogels, …, Gerstner, Science, 2011

[6] Zenke, Agnes and Gerstner, Nat. Com., 2015

[7] Sanzeni, Histed and Brunel, Phys. Rev. X, 2023

[8] Shadlen and Newsome, Cur. Op. in Neurob., 1994

[9] Mainen and Sejnowski, Science, 1995

[10] Song, …, Chklovskii, Plos Bio., 2005

[11] Okun and Lampl, Nat. Neur., 2008

[12] Pillow, …, Simoncelli, Nature, 2008

[13] Cohen and Kohn, Nat. Neur., 2011

[14] Prinz, Bucher and Marder, Nat. Neur., 2004

---

### Decision · Program_Chairs · 2023-09-21

**Decision:**

Accept (poster)

**Comment:**

The authors propose a simulation-based method to filter synaptic plasticity rules within biological constraints in large recurrent networks of spiking neurons.

The reviewers agree that the manuscript is well-written and technically sound. In general, the investigation of synaptic plasticity rules is a key problem in (computational) neuroscience. The reviewers agree that this work provides a strong contribution to the current state in this research. The manuscript derives interesting conclusions, also when comparing obtained rules with theoretically grounded mean-field approximations. Conclusions are well-supported by the simulation results.

One potential weakness is that the uncovered rules do not perform interesting functions beyond stabilization of networks. An extension in this direction would be interesting future work.

In summary, an interesting and well-performed study.